# Deriving boundary layer height from aerosol lidar using machine learning: KABL and ADABL algorithms

Thomas Rieutord[1], Sylvain Aubert[2], and Tiago Machado[1,2]

[1]Centre National de Recherches Meteorologiques, Université de Toulouse, Météo-France, CNRS, Toulouse, France
[2]Direction des Systèmes d'Observation, Météo-France, Toulouse, France

**Correspondence:** Thomas RIEUTORD (thomas.rieutord@meteo.fr)

**Abstract.** The atmospheric boundary layer height (BLH) is a key parameter for many meteorological applications, including air quality forecasts. Several algorithms have been proposed to automatically estimate BLH from lidar backscatter profiles. However recent advances in computing have enabled new approaches using machine learning that are seemingly well suited to this problem. Machine learning can handle complex classification problems and can be trained by a human expert. This paper describes and compares two machine-learning methods, the K-means unsupervised algorithm and the AdaBoost supervised algorithm to derive BLH from lidar backscatter profiles. The K-means for Atmospheric Boundary Layer (KABL) and AdaBoost for Atmospheric Boundary Layer (ADABL) algorithm codes used in this study are free and open source. Both methods were compared to reference BLHs derived from colocated radiosonde data over a two-year period (2017–2018) at two Météo-France operational network sites (Trappes and Brest). A large discrepancy in the root-mean-square error (RMSE) and correlation with radiosondes was observed between the two sites. At the Trappes site, KABL and ADABL outperformed the manufacturer's algorithm, while the performance was clearly reversed at the Brest site. We conclude that ADABL is a promising algorithm (RMSE of 550 m at Trappes, 800 m for manufacturer) but has training issues that need to be resolved; KABL has a lower performance (RMSE of 800 m at Trappes) than ADABL but is much more versatile.

## 1 Introduction

The atmospheric boundary layer is the lowest part of the troposphere and is the region that is directly influenced by surface forcings. It is the layer within which most human activities take place, and all pollutants emitted at ground level are dispersed within this layer. The key parameter used to model this dilution is the depth of this layer, i.e., the boundary layer height (BLH). Because BLH can vary from a few tens of meters to approximately 2 km within a single day, the volume available for the dilution of pollutants can vary considerably and is a crucial parameter for reliable warnings of poor air quality (Stull, 1988; Dupont et al., 2016). However, BLH is one of the largest sources of uncertainty in air quality models (Mohan et al., 2011) and there is a need to better evaluate this parameter (Arciszewska and McClatchey, 2001). Accurate representation of the physical processes within the boundary layer are also important for numerical weather prediction models (Seity et al., 2011). In the study of physical processes in the boundary layer, with large eddy simulations (Lenschow et al., 2012) or with measurements

(Brilouet et al., 2017), BLH is often used as a normalization of the vertical profiles. Therefore, it is important to compare BLH calculated in models with that derived from measurements.

However, measuring BLH is not straightforward. As stated in Seibert et al. (2000), there are no systems that meet all of the requirements for making reliable BLH estimates. The best estimate of BLH can be achieved via the synergistic use of multiple instruments, but, adding instruments limits the number of sites where estimates can be made. In this paper, we focus on a single instrument, aerosol lidar (see Sect. 2.1.1 for more information), that is already widely used (Haeffelin et al., 2012). Aerosol lidars are active remote sensing instruments that emit a laser pulse into the atmosphere and measure the amount of light backscattered from aerosols as a function of the vertical range from the instrument. Because aerosols are more concentrated in the boundary layer than in the overlying free troposphere, there is often a sharp decrease in the backscatter profile between these two layers. However, this decrease can be blurred or perturbed by other strong signals (e.g., clouds, aerosol residing in elevated or residual layers, and small-scale structures) and instrumental noise. For these reasons, numerous studies exist concerning the derivation of BLH from aerosol lidar. Melfi et al. (1985) use a simple thresholding of the signal. Other methods are based on calculations of the derivative function of the backscatter profile. For example, Hayden et al. (1997) take the minimum of the gradient, Menut et al. (1999) use the height where the second derivative is zero (the inflection point) as well as the variance of the signal, and Senff et al. (1996) use the derivative of the logarithm of the backscattered signal. One of the most used methods is the wavelet covariance transform, which searches for the maximum in the convolution between the backscatter profile and a Haar wavelet (Gamage and Hagelberg, 1993; Cohn and Angevine, 2000; Brooks, 2003). More recent studies have been based on backscatter signal analysis such as the STRucture of the ATmosphere (STRAT) algorithm (Morille et al., 2007) and the Characterising the Atmospheric Boundary layer based on Automatic lidar and ceilometer Measurements (CABAM) algorithm (Kotthaus and Grimmond, 2018). Graph theory has also been used to impose continuity constraints (both vertically and in time) in BLH estimates, e.g., the *Pathfinder* algorithm (De Bruine et al., 2017). Inspired by image processing, some methods use Canny edge detection in addition to backscatter signal analysis (Morille et al., 2007; Haeffelin et al., 2012). An extension of *Pathfinder* including the detection of the continuous aerosol layer was made in *PathfinderTURB* (Poltera et al., 2017). These studies demonstrate that estimating BLH from aerosol lidar is still an open area of research.

In addition, artificial intelligence (AI), as a set of techniques aiming to reproduce human intelligence with machines, has reemerged in the last decade because of the simultaneous increase in the amount of available data and computational power. Both have reached levels that enable previously impractical applications. AI is capable of tackling complex classification problems, especially in image classification (Krizhevsky et al., 2012). Such breakthroughs were made possible by deep convolutional neural networks (LeCun et al., 2015); however, AI encompasses many other techniques that also benefit from larger datasets and increased computational power (Besse et al., 2018). In this paper, we explore how the estimation of BLH from backscatter profiles can be formulated as a classification problem and how appropriate algorithms can be applied to solve this problem. Machine-learning techniques are categorized into two broad families: supervised learning (mimicking a reliable reference) and unsupervised learning (learning without a reference; Hastie et al., 2009). Toledo et al. (2014) have already described a method that falls within the scope of AI. They used unsupervised learning to classify whether measurement points were within the boundary layer. This method has yielded convincing results in previous studies (Toledo et al., 2017; Caicedo

et al., 2017; Rieutord et al., 2014) and is pursued here using the K-means for Atmospheric Boundary Layer (KABL) algorithm. KABL has been extensively tested and is shared via an open-source code. In addition, we test an alternative adaptive boosting (AdaBoost) machine-learning algorithm, the AdaBoost for Atmospheric Boundary Layer (ADABL) algorithm. Both algorithms classify whether the measurement points are inside or outside of the boundary layer; however, ADABL learns the characteristics of both groups from a training set. The training set consists of atmospheric boundary layer identifications made by human experts, which is acknowledged as being more reliable than available automatic methods (Seibert et al., 2000). Algorithms classifying from a reference dataset (e.g., ADABL) are called supervised algorithms, while algorithms classifying without a reference dataset (e.g., KABL) are called unsupervised algorithms. Supervised algorithms make it possible to automatically reproduce human expertise in boundary layer identification. To our knowledge, this is the first time that a supervised algorithm has been applied to this problem. This study is of practical interest because it includes the publication of the source code, which only uses free software.

In Sect. 2, we describe the data used in this study, i.e., the lidar data in the algorithm inputs, reference radiosonde data, and ancillary data used to sort the meteorological conditions. In Sect. 3, we describe the two machine-learning algorithms (KABL and ADABL) and the procedures used to evaluate them. In Sect. 4, we present the results of our study, which consists of a sensitivity analysis of the KABL algorithm, a comparison of the methods with the radiosonde data over a two-year period, and a case study. In Sect. 5, we discuss the results, limitations, and prospects of our study. The final section is dedicated to the conclusions that can be drawn from our study.

## 2 Material

Our study used data from the Météo-France operational network. We used colocated radiosonde and aerosol lidar data over two sites: Brest (a coastal city in north-western region of France) and Trappes (an inland suburban area within Paris). The dataset spanned two years: 2017 and 2018. A case study was conducted on August 2, 2018, for the Trappes site.

### 2.1 Lidar data

#### 2.1.1 Lidar network

Since 2016, Météo-France has deployed a network of six automatic backscatter lidars to help the Volcanic Ash Advisory Center of Toulouse characterize layers of volcanic ash and aerosol in the atmosphere. One of the six sensors can be quickly redeployed at a more suitable geographic location depending on the transport event being tracked. The network, fully operational since April 2017, is continuously functioning and has detected aerosol events at altitudes of up to 17 km. It is part of the wider automatic lidar and ceilometer network of the E-PROFILE program described in Haefele et al. (2016).

Two sampling sites in this network were selected: Brest (48.444° N, 4.412° W, 94 m above sea level) and Trappes (48.773° N, 2.0124° E, 166 m above sea level). Both sites are equipped with a Mini Micro Pulse LiDAR (MiniMPL), built by Sigma Space Corporation with an exterior casing provided by Envicontrol. A MiniMPL unit from the Météo-France network is shown

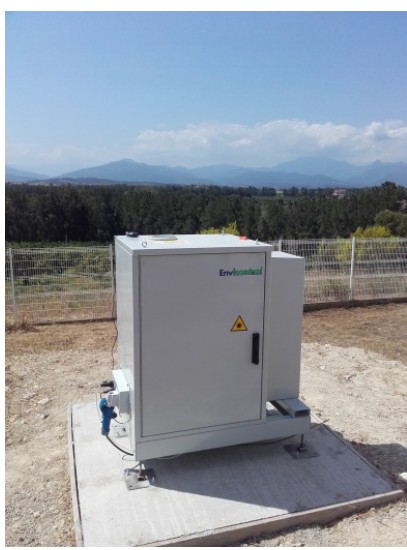

**Figure 1.** Mini Micro Pulse LiDAR (MiniMPL) unit from the Météo-France network.

in Figure 1. MiniMPL is a compact version of the Micro Pulse LiDAR (MPL) systems approved for the global NASA Micro-Pulse Lidar Network (MPLNET). A comprehensive description of MiniMPL can be found in Ware et al. (2016).

### 2.1.2 Data processing

MiniMPL acquires profiles of atmospheric backscattering at high frequency (2500 Hz) using a low-energy pulse (3.5 $\mu$J) emitted by a Nd:YAG laser at 532 nm. The profiles are acquired in photon-counting mode and, in our present configuration,
averaged over 5 min and 30-m vertical resolution bins. The instrument uses a monostatic coaxial design where the laser beam and the receiver optics share the same axis. Because of geometrical limitations, only a fraction of the signal can be recovered in the near field. Therefore, in our system, the first usable data are available at 120 m above ground level.

The instrument has polarization capability, collecting backscattered photons in two channels with the measured raw signals in the "copolarized" and "cross-polarized" channels suffixed $co$ and $cr$, respectively (the intrument uses both circular and linear
depolarization; see Flynn et al. (2007), for more details). These raw signals are processed to obtain the quantity of interest, i.e., the range-corrected signal ($RCS$), which is also called the normalized relative backscatter. This processing consists of several procedures including background, overlap, afterpulse, and dead-time corrections. A comprehensive description of the processing is given in Campbell et al. (2002). The "copolarized" and "cross-polarized" range-corrected signals, $RCS_{co}$ and $RCS_{cr}$, respectively, as delivered by the manufacturer's software, are used as predictors for the machine-learning algorithms
described in Sect. 3.

The raw data type and format depends on the instrumental device used. To make the algorithms usable on other devices, we converted the files to a harmonised format using the *raw2l1* software [1]; we then used these files as the algorithm input.

## 2.2 Radiosonde data

The algorithms were evaluated with respect to estimates derived from radiosonde (RS) profiles. Météo-France operates several RS sites for the World Meteorological Organization Global Observing System. Two RS sites are colocated with the lidars at Brest and Trappes. These sites are equipped with Meteomodem robotsondes and typically launch a Meteomodem M10 sonde at 11:15 UTC and 23:15 UTC every day.

Many methods exist to estimate BLH from RS data, several of which have been used in the literature. Some of these methods are listed below.

- Parcel method: BLH is the height at which the profile of the potential temperature $\theta$ reaches its ground value.

- Humidity gradient method: BLH is the height at which the gradient of the relative humidity is strongly negative.

- Bulk Richardson number method: BLH is the height at which the bulk Richardson number exceeds 0.25 (this threshold varies among studies).

- Surface-based inversion: BLH is the height at which the gradient temperature profile reaches zero.

- Stable layer inversion: BLH is the height at which the gradient of the potential temperature profile reaches zero.

Hennemuth and Lammert (2006) used the parcel and humidity gradient methods. Collaud Coen et al. (2014) used all the techniques mentioned above and recommend the bulk Richardson number method for all cases. Guo et al. (2016) used the bulk Richardson number for a two-year climatology. Seidel et al. (2010) compared the parcel, humidity gradient, and surface-based inversion methods, as well as other methods, over a period of 10 years at 505 sites worldwide. Seidel et al. (2012) compared several methods and recommend the bulk Richardson number method.

Following the recommendations in Figure 10 of Seibert et al. (2000), we chose to compute BLH using the parcel method for the 11:15 UTC sounding and the bulk Richardson number for the 23:15 UTC sounding and refer to this estimate as BLH-RS from now on.

## 2.3 Ancillary data

Ancillary data were used to describe the meteorological situation at the observation sites. These data were not used by the machine-learning algorithms. All the instruments were colocated with the lidar and radiosonde launches.

- Rain gauges were used to detect rain events.

---

[1] *raw2l1*, which is maintained by the Site Instrumental de Recherche par Télédétection Atmosphérique and is publicly available at https://gitlab.in2p3.fr/ipsl/sirta/raw2l1

- Vaisala Ceilometer CL31 instruments were used to detect the cloud base height and distinguish cases with clouds on top of, or inside, the boundary layer. Even though MiniMPL is capable of detecting clouds, we relied on the CL31 cloud detection because the MiniMPL algorithm was found to report non-existent clouds.

- Scatterometers were used to estimate the horizontal visibility and detect the occurrence of fog.

## 3 Machine-learning methods

### 3.1 Supervised learning method

Supervised methods learn from a reference. Such methods are divided into two families: classification, which aims to find the frontiers between groups, and regression, which aims to approximate a function. In this study, we treat the BLH estimation as a classification problem where we wish to classify the lidar measurement at each range gate as belonging to either 'boundary layer' or 'free atmosphere'. Then, the highest point of the 'boundary layer' class indicates the BLH estimate. Several supervised algorithms were compared to maximize accuracy (see Sect. 3.1.3), here, we describe Adaboost, which was the algorithm selected for this study. Boosting algorithms are a very powerful family of algorithms that were developed for classification but can also be used for regression (Hastie et al., 2009). In particular, the AdaBoost algorithm is designed for binary classification (Freund and Schapire, 1997) and is therefore well suited to our problem.

### 3.1.1 AdaBoost algorithm

Let us consider the following problem. We have $N$ vectors $x_i \in \mathbb{R}^p$ (here, the numbers of predictors, $p = 4$: seconds since midnight, height above ground, copolarized channel and cross-polarized channel), and for each vector, we have a binary indicator $y_i \in \{-1, 1\}$ (−1 for 'boundary layer', 1 for 'free atmosphere'). From the sample $(x_i, y_i)_{i \in [\![1,N]\!]}$, where $[\![1, N]\!]$ is the ensemble of integers from 1 to $N$, we want to predict the output indicator $y_{new}$ of any new vector $x_{new}$. To do so, we must find a rule based on the $x_{new}$ coordinate values (the features) to cast it into the appropriate class. Decision tree classifiers (Breiman et al., 1984) perform this casting one feature at a time. For example, in Figure 2, there are black and white points in a two-dimensional space. The black points are mostly located where $X_1$ is low, hence the rule "if $X_1 < t_1$, then the point is black." However, in the other region, where $X_1 > t_1$, there are still some black points, all with low $X_2$. Therefore, we add the rule "if $X_2 < t_2$, then the point is black, else it is white." Decision trees are classifiers made up of such "if" statements with various depths and thresholds. The deeper the tree, the more accurate the border but the more complex the decision and the longer it takes to train. Deep trees are strongly subject to overfitting and are less efficient than other methods. However, shallow decision trees are valuable because of their simplicity and their speed, even though their performance are quite limited (Hastie et al., 2009). They are often used as *weak learners*, that is, classifiers with poor performance (but better than random) that are very simple (Freund and Schapire, 1997). In this study, weak learners in AdaBoost are trees with a maximum depth of five (a maximum of five forks between the root and the leaves).

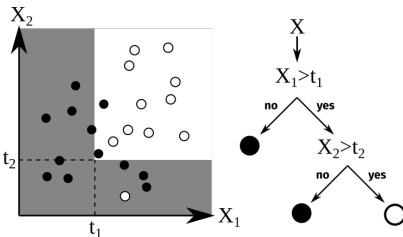

**Figure 2.** Illustration of binary classification with decision trees on two-dimensional artificial data.

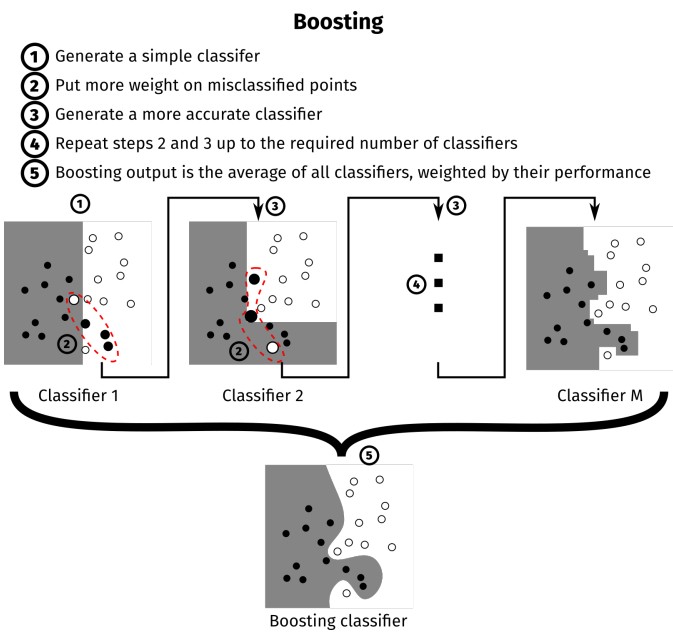

**Figure 3.** Illustration of boosting on two-dimensional artificial data with two classes.

AdaBoost is based on decision tree classifiers. It aggregates these classifiers to determine the most accurate border. The concept behind AdaBoost is illustrated in Figure 3. First, a shallow decision tree is fitted to the entire dataset using the Classi-

fication and Regression Tree (CART) algorithm (Hastie et al., 2009). All points have the same weight in this first step. Some points in the dataset are misclassified, and the error of the classifier is the weighted average of the misclassified points. Another shallow decision tree is then fitted on a resampled dataset where the previously misclassified points are over-represented. This new tree has new misclassified points that will be over-represented in the training of the next tree, and so on, up to the specified number of trees ($M = 200$ in our case). The detailed algorithm is described in Hastie et al. (2009), algorithm 10.1, and in

Schapire (2013).

### 3.1.2 Training of the algorithm

Such an algorithm needs to be trained using a trustworthy reference. On days where the boundary layer is easily visible to a human expert, the top of the boundary layer can be drawn by hand; all points below this limit are in the class 'boundary layer' and all points above this limit are in the class 'free atmosphere'.

In this study, two days were classified by hand. These two days where chosen because the boundary layers on these days were easily visible; the two hand-classified days were at different sites in different seasons. The first hand-classified day was a clear summer day in Trappes, shown in Figure 4 (left); a stable boundary layer is present near the ground during the night, topped by a residual layer and a few clouds between 02:00 UTC and 04:00 UTC. A mixed layer started to develop at 09:00 UTC and remained at approximately 2000 m for the rest of the day. At approximately 22:00 UTC, a new stable layer appeared to develop near the ground; however, it is not very clear where this layer started or what its extent was. The second hand-classified day was a clear winter day in Brest, shown in Figure 4 (right): a stable boundary layer was present near the ground during the night, topped by a residual layer, which was shallower than the layer observed at the Trappes site. The mixed layer started to develop at 08:00 UTC and remained at approximately 1000 m with the height of the layer gradually decreasing throughout the day. At approximately 17:00 UTC, aerosols appeared to accumulate in a thin layer close to the ground; therefore, we chose to select the top of this thin layer as BLH.

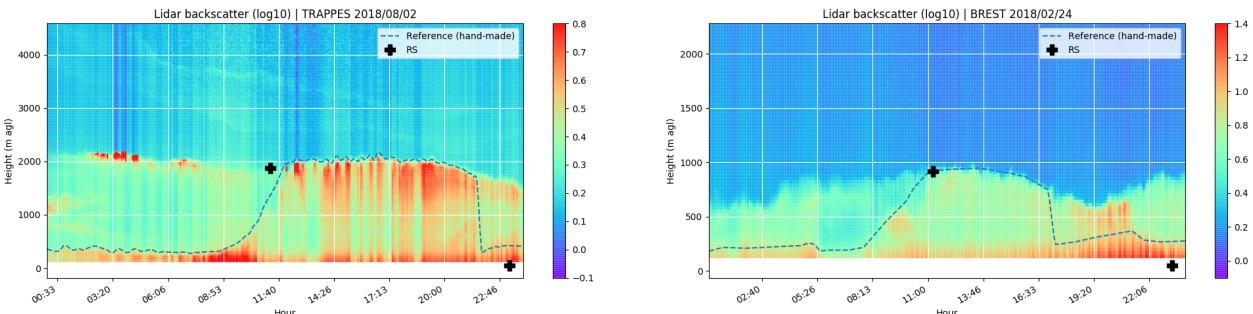

**Figure 4.** Hand-drawn reference classification and radiosonde estimates overlaying the lidar range-corrected signal for two days: August 2, 2018, at the Trappes site (left) and February 24, 2018, at the Brest site (right).

The coordinates of the points on the hand-drawn BLHs were obtained using the Visual Geometry Group Image Annotator software[2]. Then, the output curves were interpolated with a cubic spline to match the lidar temporal resolution. Given the resolution of the lidar, this method of labeling the data results in $N = 86,400$ individuals in total (this number is the product of 288 profiles per day, 150 vertical range bins and two labeled days).

---

[2]Publicly available online at https://www.robots.ox.ac.uk/~vgg/software/via/via-1.0.6.html.

 ### 3.1.3 Retained configuration

Four predictors were used: the two lidar channels, time (number of seconds since midnight), and altitude (meters above ground level). The ADABL configuration used was

- – weak learner: decision tree of depth five;

- – number of weak learners: 200; and

195 – predictors: time, altitude, $RCS_{co}$, and $RCS_{cr}$.

This configuration was chosen because more complex classifiers do not necessarily improve the performance. The computation time of the algorithm was still reasonable: training took 23 s on the full dataset and predicting BLH for a full day took 3.7 s with a modern laptop. AdaBoost was chosen after a comparison of multiple classification algorithms, i.e., random forest, nearest neighbor, decision trees, and label spreading (study not shown here). The benchmark score was the accuracy as measured 200 by the percentage of individuals that were correctly classified. The accuracy was estimated by group K-fold cross-validation, where labeled datasets are grouped into chunks of three consecutive hours, one group was used as a testing set and all the rest as a training set. This operation was repeated until each group was used as the testing set. The resulting accuracy was 96%. However, this figure overestimates the generalization ability of AdaBoost. A more correct estimation would be obtained with an independent validation set (e.g., a new hand-classified day). An independent validation set was not used here because the 205 cross-validation accuracy was only used to discriminate between the classification algorithms.

It is possible to quantify the relative importance of the predictors (Breiman et al., 1984; Hastie et al., 2009). After training, the relative importance of the time, $RCS_{co}$, $RCS_{cr}$, and altitude predictors was 30.3%, 28.4%, 26.5%, and 14.8%, respectively.

## 3.2 Unsupervised learning methods

Unsupervised methods aim to identify groups in the data. In our case, we want to identify the group 'boundary layer'. The 210 BLH estimate is then the upper boundary of this group. Two unsupervised learning algorithms were tested: K-means and expectation–maximization (EM).

### 3.2.1 K-means algorithm

The K-means algorithm is a well proven and commonly used algorithm for data segmentation (Jain et al., 1999; Pollard et al., 1981) and consists of three steps, where $K$ is the number of clusters specified by the user.

1. Initialization: $K$ centroids $m_1, ..., m_K$ are initialized at random locations inside the feature space.

2. Attribution: The distances from all points to all centroids $(d(x_i, m_k))_{k \in [\![1,K]\!], i \in [\![1,N]\!]}$ are computed, and points are attributed to the closest centroid:

$C(i) = arg \min_k \{d(x_i, m_k)\}$.

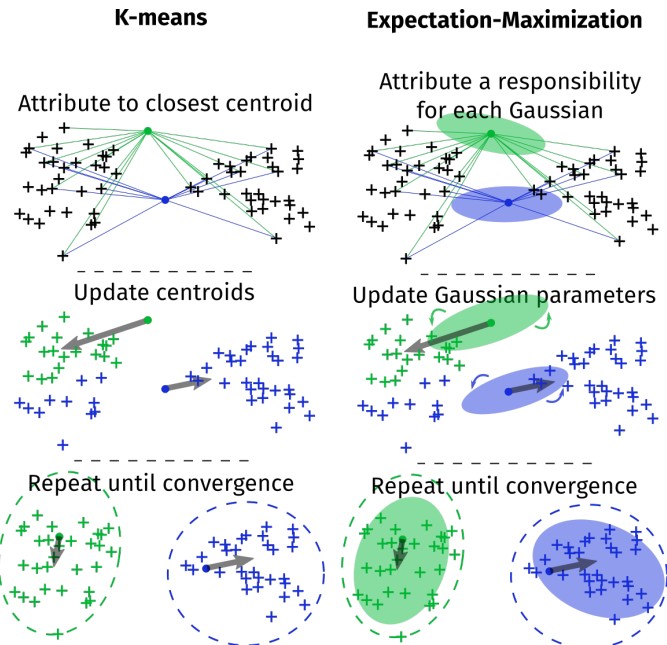

**K-means**

Attribute to closest centroid

Update centroids

Repeat until convergence

**Expectation-Maximization**

Attribute a responsibility for each Gaussian

Update Gaussian parameters

Repeat until convergence

**Figure 5.** Illustration of the K-means and expectation–maximization algorithm on two-dimensional artificial data with two clusters.

3.  Update: The centroids are re-defined as the average point of the cluster: $m_k = \frac{\sum_{i=1}^{N} x_i \mathbf{1}_{C(i)=k}}{\sum_{i=1}^{N} \mathbf{1}_{C(i)=k}}$.

Steps 2 and 3 are repeated until the centroids stop moving. It has been shown that this algorithm converges to a local minimum of the intra-cluster variance (Selim and Ismail, 1984). Figure 5 (left) illustrates this algorithm.

### 3.2.2 EM algorithm

The EM algorithm assumes that each group $k \in [\![1, K]\!]$ is generated by a Gaussian distribution $(\mu_k, \Sigma_k)$. The algorithm iteratively estimates the parameters $\hat{\mu}_k$, $\hat{\Sigma}_k$, and the *responsibility* for each Gaussian $\hat{\gamma}_k^i$, where the *responsibility* is the probability of the point $x^i$ being generated by the $k$-th Gaussian. Points are then attributed to the group with the highest responsibility: $C(i) = arg \max_k(\hat{\gamma}_1^i, ..., \hat{\gamma}_K^i)$. Figure 5 (right) illustrates this algorithm.

The K-means and EM algorithms are very similar. If we assume that all Gaussians distributions have the same fixed variance and that this variance tends to zero, the EM and K-means algorithms are the same. However, K-means does not rely on a Gaussian assumption.

### 3.3 Flowchart and description of KABL parameters

A simplified flowchart of KABL and ADABL is shown in Figure 6. This section focuses on the KABL parameters to introduce the sensitivity analysis made in Sect. 4.1. The parameters of the KABL software are detailed here.

– **algo:** The applied machine-learning algorithm. Possible values are:

- – 'gmm' for the EM algorithm (Gaussian mixture model); and

- – 'kmeans' for the K-means algorithm.

- **classif_score:** The internal score used to automatically choose the number of clusters (only used when n_clusters = 'auto'). See Sect. 3.4 and Table 1 for a description of the internal scores.

- **init:** Initialization strategy for both algorithms. Three choices are available:

  - – 'random': randomly pick an individual as the starting point (both K-means and EM);

  - – 'advanced': use a more sophisticated initialization (kmeans++ for K-means (Arthur and Vassilvitskii, 2007) and the output a K-means pass for EM); and

  - – 'given': start at explicitly selected point coordinates.

- **max_height:** The height (meters above ground level) at which the profiles are cut.

- **n_clusters:** The number of clusters to be formed (between two and six). This is either explicitly given or determined automatically to optimize the score given in classif_score.

- **n_inits:** The number of repetitions of the algorithm. When this number is larger, the algorithm is more likely to find the global optimum but requires more time.

- **n_profiles:** The number of profiles concatenated prior to the application of the algorithm. For example, if n_profiles = 1, only the current profile is used. If n_profiles = 3, the current profile and the two previous profiles are concatenated and input into the algorithm.

- **predictors:** The list of variables used in the classification. These variables can be different at night and during the day. For both time periods, the variables can be chosen from

  - – $RCS_{co}$: the copolarized range-corrected backscatter signal; and

  - – $RCS_{cr}$: the cross-polarized range-corrected backscatter signal.

The parameters of the KABL software are highlighted in bold in the following explanation of the KABL algorithm. A netCDF file generated by the *raw2l1* software needs to be provided as input data to KABL. The data, namely, the altitude vector $z$ (size $N_z$), the time vector $t$ (size $N_t$), and the range-corrected signals $RCS_{co}$ and $RCS_{cr}$ ($N_t \times N_z$ matrices), are extracted from this file. Such data are prepared to fulfil the machine-learning algorithm requirements. For each time, the **n_profiles** last profiles are extracted. Then, the data they contain are normalized (by removing the mean and dividing by the standard deviation); this provides a matrix $X$ ($N \times p$, where $N =$**n_profiles**$\cdot N_z$ and $p = |$**predictors**$|$ is the number of elements in the list). The matrix $X$ is the usual input for a machine-learning algorithm; it has one line for each individual observation and one column for each variable (or predictor) observed. For the BLH retrieval, the preparation also provides a vector $Z$ (size $N$) containing the altitude of each individual observation. The algorithm (either K-means or EM, as specified by **algo**) is applied

to the matrix $X$, with the parameters **n_clusters**, **init**, and **n_inits**. This results in a vector of *labels* (size $N$) that contains the cluster attribution of each individual. Finally, since by definition the boundary layer is the layer directly influenced by the ground, we look for the first change in the cluster attribution, starting from the ground level. This gives us the value of BLH for this profile. These operations are repeated until reaching the end of the netCDF file.

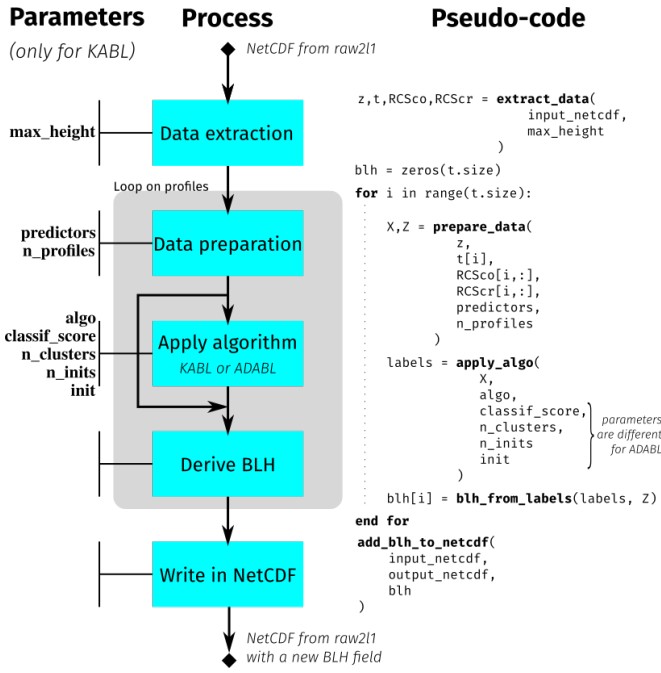

**Figure 6.** Simplified flowchart of the K-means for Atmospheric Boundary Layer (KABL) and AdaBoost for Atmospheric Boundary Layer (ADABL) algorithms with a focus on the KABL parameters. The parameters are described in Sect. 3.3 and in Table 2.

## 3.4 Performance metrics

Two types of metrics were used.

– External scores: These metrics compare the result to a trustworthy reference. They have the advantage of providing a meaningful evaluation of the performance but depend strongly on the quality of the reference (i.e., its accuracy and availability).

– Internal scores: These metrics rate how well the classification performs based only on the distances between points. They have the advantage of being always computable but are not linked to any physical property and therefore are not always meaningful.

None of these metrics are perfect; however, the information they provide allows a broader understanding of the algorithm performance.

### 3.4.1 External scores

External scores use a reference to assess the quality of the result. In our case, the reference are BLH-RS (RS-derived BLH)
and, when available, the human expert hand-classified BLH. Two external scores are used in this study. If we denote $\hat{Z}$ as the
estimated BLH (by any of the previously introduced algorithms) and $Z_{ref}$ as the reference, the external scores are as follows:

- RMSE, where lower values are better:

$$E_2 = \sqrt{\mathbb{E}\left[(\hat{Z} - Z_{ref})^2\right]}. \tag{1}$$

- The Pearson correlation, where higher values are better:

$$\rho = \frac{cov(\hat{Z}, Z_{ref})}{\sigma(\hat{Z})\sigma(Z_{ref})}. \tag{2}$$

Here, $\hat{Z}$ and $Z_{ref}$ are random variables, $\mathbb{E}[\cdot]$ denotes the mathematical expectation, and $\sigma(\cdot)$ denotes the standard deviation.
When these scores are estimated, the random variables are replaced by a sample vector and the expectation and standard
deviation are replaced by their usual estimators.

### 3.4.2 Internal scores

The quality of a classification can be quantified using scores that are based only on the labels and the distances between
points. Such scores estimate how trustworthy an estimation is without any external input. Many such scores exist with different
formulations and different strengths and weaknesses (Desgraupes, 2013). In this study, three internal scores were used:

- The silhouette score (Rousseeuw, 1987):

$$S_{sil} = \frac{b - a}{\max(a, b)}, \tag{3}$$

where $a$ is the average distance to its own group and $b$ is the average distance to the neighboring group. $S_{sil} = 1$ is the
best classification, $S_{sil} = 0$ is neutral, and $S_{sil} = -1$ is the worst classification.

- The Calinski–Harabasz index (Caliński and Harabasz, 1974):

$$S_{ch} = \frac{(N - K)B}{(K - 1)\sum_{k=1}^{K} W_k}, \tag{4}$$

where $N$ is the number of points, $K$ is the number of clusters, $B$ is the between-cluster dispersion, and $W_k$ is the
within-cluster dispersion of the cluster $k$. Higher $S_{ch}$ indicates a better classification.

- The Davies–Bouldin index (Davies and Bouldin, 1979):

$$S_{db} = \max_{k' \neq k} \left( \frac{\bar{\delta}_k + \bar{\delta}_{k'}}{d(\mu_k, \mu_{k'})} \right), \tag{5}$$

where $k$ and $k'$ are the two cluster numbers, $\bar{\delta}_k$ is the average distance between points and their cluster center for the
cluster $k$, and $d(\mu_k, \mu_{k'})$ is the distance between the cluster centers $\mu_k$ and $\mu_{k'}$. Lower $S_{db}$ indicates a better classification.

**Table 1.** Table of metrics used to measure the performance of the K-means for Atmospheric Boundary Layer (KABL) algorithm. The metrics are described in detail in Sect. 3.4.

| Metric | Type | Description | Best, worst score |
|:---:|:---:|:---:|:---:|
| **corr** | External | Pearson correlation coefficient | $1, 0$ |
| **errl2** | External | Root-mean-square error (RMSE) | $0, +\infty$ |
| **s_score** | Internal | Silhouette score | $1, -1$ |
| **db_score** | Internal | Davies–Bouldin index | $0, +\infty$ |
| **ch_score** | Internal | Calinski–Harabasz index | $+\infty, 0$ |
| **chrono** | Other | Time to estimate BLH for a 24 hr period | $0, +\infty$ |
| **n_invalid** | Other | Number of invalid BLH estimates (NaN or Inf) for a 24 hr period | $0, 288$ |

These three scores were chosen to diversify the metrics and are all implemented in Scikit-learn (version $\geq$0.20).

### 3.4.3  Other metrics

In addition to the internal and external scores, the computation time and the number of invalid values (NaN or Inf) were recorded. BLH estimates of NaN or Inf can occur when all the points of the profile are assigned to the same cluster; this reflects a faulty configuration of the algorithm. Even though these metrics do not measure how well a program is performing, they are
useful to the user.

All the metrics used to measure the performance of KABL are summarized in Table 1.

## 4  Results

### 4.1  Sensitivity analysis of the KABL algorithm

A sensitivity analysis was performed on the KABL code to identify the "best" configuration. Various KABL configurations
were extensively tested on a single day: August 2, 2018, at the Trappes site, for which we have a hand-classified reference (Figure 4 (left)). The most relevant configurations were retained and tested on the two-year lidar dataset.

There are eight parameters in the KABL code (see Sect. 3.3 for their descriptions). To assess the sensitivity of KABL to these parameters, the performance metrics (given in Sect. 3.4) were calculated using the hand-classified BLH as $Z_{ref}$ and with the output of KABL as $\hat{Z}$ for different combinations of input parameters. The output metrics given in Table 1 were tested using
the input values given in Table 2. We refer to a set of values for the KABL parameters as a *configuration*. Screening all the possible values listed in Table 2 would require 3240 different configurations.

**Table 2.** Possible values for the parameters of the KABL code. The parameters are described in details in Sect. 3.3. The dependencies between parameters result in 3240 different configurations.

| Parameter | Possible values | Meaning |
|---|---|---|
| **algo** | 'kmeans' | The K-means algorithm is used |
| | 'gmm' | The EM algorithm is used (Gaussian mixture model) |
| **classif_score** | 'silh' | The silhouette score is used |
| | 'db' | The Davies–Bouldin index is used |
| | 'ch' | The Calinski–Harabasz index is used |
| **init** | 'random' | Starting points are chosen randomly |
| | 'advanced' | Starting points are chosen with a smarter strategy |
| | 'given' | Starting points are explicitly given |
| **max_height** | 3500 | The altitude above which profile data is discarded |
| | 4500 | (meters above ground level) |
| **n_clusters** | 2 | |
| | 3 | The number of clusters to be formed is explicitly |
| | 4 | passed and is always the same |
| | 5 | |
| | 'auto' | The number of clusters is automatically chosen to optimize **classif_score** |
| **n_inits** | 10 | The number of times the algorithm is repeated with |
| | 80 | different initializations (when **init** is not 'given') |
| **n_profiles** | 1 | Only the current profile is used |
| | 2 | The current profile and the previous profile are used |
| | 3 | The current profile and the two previous profiles are used |
| | 4 | The current profile and the three previous profiles are used |
| **predictors** | 'co' | The copolarized range-corrected signal is used at all times |
| | 'co/co+cr' | The copolarized range-corrected signal is used during the daytime, and both polarization channels are used independently during the nighttime |
| | 'co+cr' | Both polarization channels are used independently at all times |

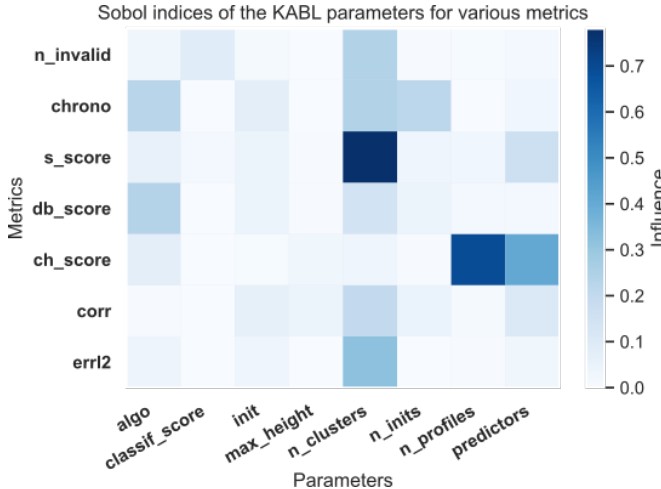

**Figure 7.** Relative influence of parameters on the different metrics. The $x$-axis indicates the parameters of the code, and the $y$-axis indicates the metrics. The shading represents the influence of the parameter on the metric with darker shading indicating a larger influence. The abbreviations for the parameters are described in Table 2 and the abbreviations for the metrics are described in Table 1.

To obtain an overview of these 3240 configurations, we started by estimating the influence of the parameters (listed in Table 2) on the different metrics (listed in Table 1). The influence of the parameters was quantified using first-order Sobol indices (Sobol, 2001; Iooss and Lemaître, 2015; Rieutord, 2017), that is, the ratio of the variance of the metric when the parameter was fixed over the total variance of the metric. If we denote $Y$ as the metric and $X$ as the vector of the parameters, where all the parameters are treated as random variables, the first-order Sobol index of the $i$-th parameter is defined as $S_i = V(\mathbb{E}[Y|X_i])/V(Y)$, where $V(\cdot)$ denotes the variance and $\mathbb{E}[\cdot]$ denotes the expectation. A higher Sobol index indicates a larger influence.

Figure 7 shows the Sobol indices obtained with the KABL computer code. Examining the matrix line by line, one can see that the different metrics are sensitive to different parameters. For example, the silhouette score is very sensitive to **n_clusters** while the Calinski–Harabasz index is sensitive to **n_profiles** and **predictors**. Examining the matrix column by column, one can see that some parameters are more influential than others (e.g., **classif_score** is much less influential than **n_clusters**). This matrix highlights the main effects of changing a parameter and, therefore, how to set each parameter appropriately. For each parameter, we examined the metrics that it influences and determined the preferred configuration.

Critical parameters are indicated in Figure 7 by the darkest blue columns, namely, **n_clusters**, **algo**, **predictors**, and **init**[3]. For each parameter, Figure 8 shows the distribution of the relevant output given the parameter value (violin plots are explained in Hintze and Nelson (1998)). For example, Figure 8a has the value of **algo** on its $x$-axis and the computing time on its $y$-axis. The 3240 different configurations were divided into two groups; those with **algo**='kmeans' and those with **algo**='gmm'.

---

[3]Even though **n_profiles** has a large Sobol index for the Calinski–Harabasz index, this influence was not explored because it is known: it is due to the linear increase in this index with the number of points.

**Table 3.** Retained values for the parameters of the KABL code after the sensitivity analysis.

| Parameter | Retained values |
|---|---|
| algo | 'kmeans' |
| classif_score | 'db' |
| init | 'given' |
| max_height | 4500 |
| n_clusters | 3 |
| n_inits | 10 |
| n_profiles | 1 |
| predictors | 'co' |

Figure 8a shows a smoothed histogram of the computing time for the divided populations. The other panels in Figure 8 were constructed in the same manner. Each line corresponds to a critical parameter, and we represent the two most influenced outputs according to Figure 7.

The parameter values were chosen to give the optimal values for the metrics they influence. The optimal values are indicated by a yellow star in each plot. To set **algo**, we examined the computing time (Figure 8a) and the Davies–Bouldin index (Figure 8b). These figures indicate that 'kmeans' is the best choice for both metrics (resulting in a lower computing time and a lower Davies–Bouldin index). To set **init**, we examined the correlation (Figure 8c) and the computing time (Figure 8d). In this case, 'given' appears to be the best choice. To set **n_clusters**, we examined RMSE (Figure 8e) and the silhouette score (Figure 8f). They indicate that the best numbers of clusters are three and 'auto', respectively. We chose to give priority to RMSE because the silhouette score has very high values for two clusters, which is suspicious given the presence of a cloud and a residual layer on this day. To set **predictors**, we examined the silhouette score (Figure 8g) and the Calinski–Harabasz index (Figure 8h); here, 'co' appears to be the best choice. Following this methodology, we can identify a few configurations worth trying. These configurations were tested on the two-year dataset. The configuration used to generate the results in Sect. 4.2.1 is given in Table 3. This configuration was chosen to maximize the correlation between KABL and RS at the Trappes site.

## 4.2 Two-year comparison

BLH estimates from the three methods (KABL, ADABL, and the manufacturer's algorithm) were compared to BLH-RS over a two-year period.

### 4.2.1 Overall comparison

As explained in Sect. 3.4, two external scores, RMSE and the correlation, were used to assess the quality of the estimates. In equations 1 and 2, the reference BLH $Z_{ref}$ was set to the BLH-RS, as described in Sect. 2.2. To compute the scores, the BLH

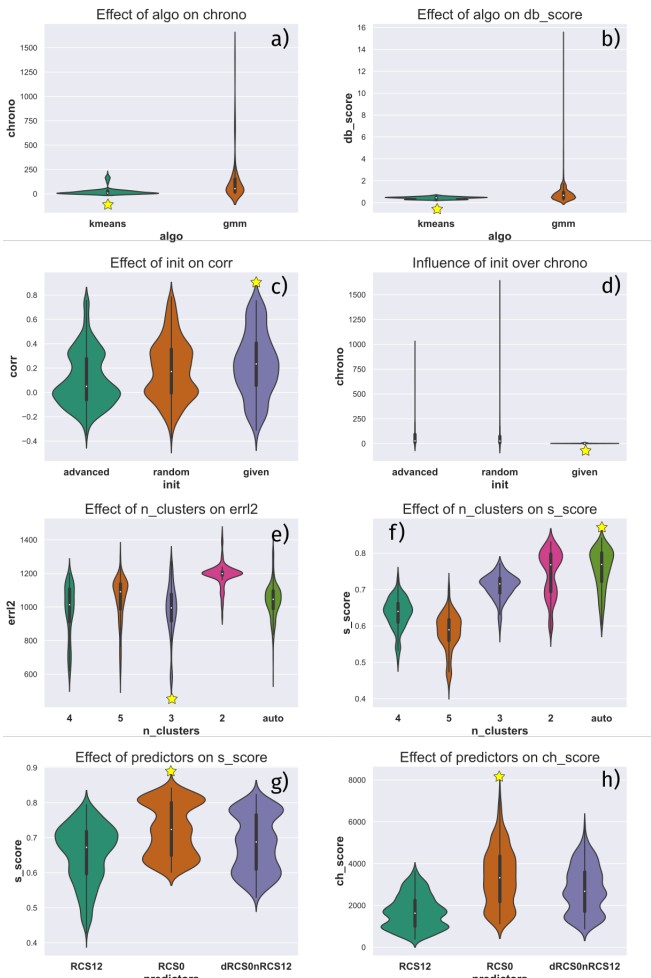

**Figure 8.** Distribution of the relevant outputs for the critical inputs. The effect of **algo** on (a) the computing time and (b) the Davies–Bouldin index. The effect of **init** on (c) the correlation and (d) the computing time. The effect of **n_clusters** on (e) the root-mean-square error (RMSE) and (f) the silhouette score. The effect of the predictors on (g) the silhouette score and (h) the Calinski–Harabasz index. For each panel, the best parameter value is highlighted by a yellow star.

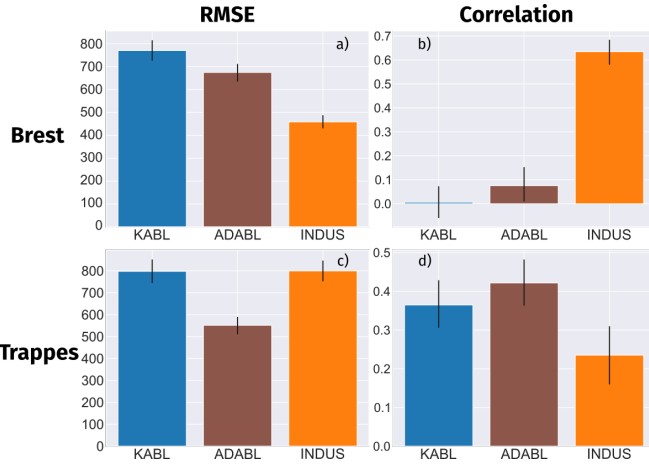

**Figure 9.** Results of a two-year comparison with the radiosonde (RS) estimates at both sites for two metrics: RMSE and correlation. INDUS refers to the manufacturer's algorithm, KABL and ADABL refer to the eponymous estimates. Cases at night or with rain, fog, an RS-estimated boundary layer height (BLH) of under 120 m, or clouds under 3000 m were removed. The 95% confidence intervals were estimated using percentile bootstrapping (Davison and Hinkley, 1997).

estimates from the lidar and radiosonde must be colocated. For each BLH-RS, the corresponding lidar BLH estimate is the

360 average of all available estimates within the 10 min following the release of the radiosonde (this translates to one or two lidar estimates). Using the ancillary measurements presented in Sect. 2.3, the following meteorological conditions were discarded:

- rain (rain gauge measures the rainfall as >0 mm);

- fog (scatterometer measures the visibility as <1000 m);

- low level cloud (ceilometer measures the cloud base height as <3000 m);

- BLH-RS is below 120 m (blind zone for lidar); and

- nighttime (RS launched at 23:15 UTC).

This selection rejects a large part of the dataset but ensures that only well-defined cases are retained for the comparison. In total, 178 RS measurements from Trappes and 101 RS measurements from Brest were used for the overall comparison.

Figure 9 displays how the three methods, KABL (blue bars), ADABL (brown bars) and manufacturer (orange bars), compare

to BLH-RS. The first column represents RMSE $E_2$ (lower is better), and the second column represents the correlation $\rho$ (higher is better). The upper row shows the results for the Brest site, the lower row shows the results for the Trappes site. While both KABL and ADABL outperform the manufacturer's algorithm at the Trappes site, neither algorithm does at the Brest site. While the correlation for both KABL and ADABL is higher than that for the manufacturer's algorithm at the Trappes site, it collapses to close to zero for KABL at the Brest site (0.07 for ADABL). The RMSE values can be compared to the values given in

Haeffelin et al. (2012). For KABL, we find 770 m at the Brest site and 798 m at the Trappes site, while for ADABL, we find

675 m at the Brest site and 552 m at the Trappes site. Our values are notably higher than those in Haeffelin et al. (2012). This is likely due to the larger extent of our dataset (178 RS at Trappes and 101 at Brest, spanning a two-year period) and the low maturity of the algorithms. ADABL has better correlation and RMSE values than KABL at both sites. The manufacturer's algorithm performs well without any specific tuning on our part. It uses a wavelet covariance transform, as described in Brooks (2003). This result is not surprising because the wavelet method has been shown to be robust in numerous studies, especially in Caicedo et al. (2017), who included a cluster analysis method and concluded that the wavelet method is preferred.

### 4.2.2   Seasonal and diurnal cycles

To quantify the ability of the algorithms to provide a consistent BLH estimate, Figure 10 shows the seasonal cycle (monthly average) and the diurnal cycle (six-minute average) at both sites. For each estimator, the thick line represents the average BLH estimate and the shaded area represents the inter-quartile gap. Rain, fog, and low-cloud conditions were discarded. For the monthly average, the night-time values were also removed. The seasonal cycle is reversed when only night-time values are studied, with BLH-RS being lower in summer than winter, on average. For other estimators, we do not see such a difference between the day and night seasonal cycles (not shown).

At the Brest site (Figure 10a), estimates made by the manufacturer's algorithm are lower than those made by KABL and ADABL and estimates made by ADABL are usually higher than those made by KABL (except in July). BLH-RS values were low in summer (June–October), high in February and March (higher than the KABL estimates), and between the manufacturer's and KABL estimates during the rest of the year. Overall, the manufacturer's algorithm displays the seasonal cycle that is closest to BLH-RS, while KABL and ADABL both overestimate BLH. The inter-quartile range (shaded areas) is large for all estimates.

At the Trappes site (Figure 10c), KABL and ADABL also overestimate BLH in comparison to the BLH-RS, while the manufacturer's estimate is close. The seasonal cycle is more visible at Trappes than at Brest, and all BLH estimates are higher in summer than in winter. The most pronounced cycle is given by KABL, while the least pronounced cycle is given by BLH-RS. The inter-quartile range are also very large, especially in summer, reflecting the variation of BLH between day and night.

Figures 10b and 10d show the diurnal cycle, where all values within the same six-minute period in the day were averaged. Because the radiosondes are only launched twice a day, at 11:15 UTC and 23:15 UTC, an equivalent BLH-RS diurnal cycle cannot be drawn. However, we used the average and quartile values at these times as checkpoints for the other estimates. The manufacturer's and KABL estimates both have very smooth diurnal cycles, with lower BLH at night and maximum BLH around 15:00 UTC at the Trappes site and around 13:00 UTC at the Brest site. The KABL average is always higher than that calculated by the manufacturer's algorithm. The ADABL estimation has a very different diurnal cycle, similar to the conceptual image we have of the boundary layer. Indeed, ADABL was trained using hand-classified BLHs that reflect this conceptual image. Therefore, it is not surprising that ADABL reproduces this image well; however, it may fail to adapt to special cases. It appears that the "time" predictor (the number of seconds since midnight) has a large influence that is not balanced by the other predictors. This is likely because ADABL was trained on only two days, resulting in an unbalanced importance for sunrise and sunset on these particular days and at these locations. To balance this importance, the AdaBoost algorithm needs to be trained on more days and at more sites with a representative selection of cases.

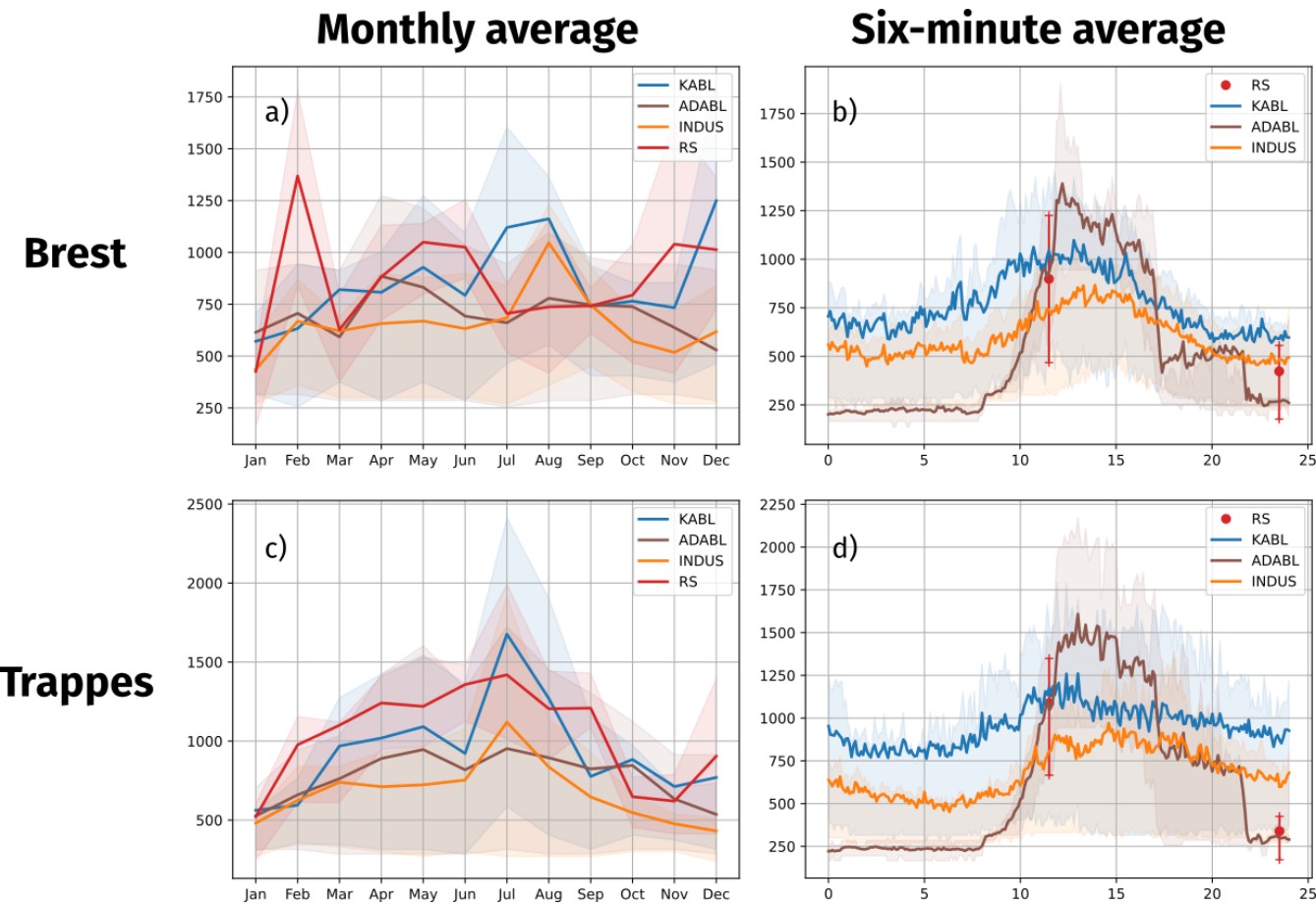

**Figure 10.** (a, c) Seasonal and (b, d) diurnal cycles of all BLH estimates at both sites. INDUS indicates the manufacturer's algorithm. Thick lines represent the average, and the shaded area represents the quartiles.

## 4.3 Case study

The chosen case study was for April 19, 2017, at the Trappes site. The boundary layer was clearly visible and had nearly all the features of the conceptual image. The case study was for a day that was not included in the ADABL training set, so as not to bias the comparison in favour of ADABL.

Figure 11 represents the range-corrected copolarized backscatter signal ($RCS_{co}$) in shaded colors. The $x$-axis indicates the hour of the day (UTC), and the $y$-axis indicates the height (meters above ground level). The different BLH estimates are represented by dotted lines: blue indicates KABL, orange indicates the manufacturer's algorithm, and green indicates ADABL. At the beginning of the day, there is a thick residual layer containing some plumes. Both KABL and the manufacturer's algorithm include these plumes in the boundary layer. Conversely, ADABL gives a very low estimate where there is no visible frontier. In the morning (from 08:00 UTC to 12:00 UTC), all the algorithms capture the morning transition reasonably well.

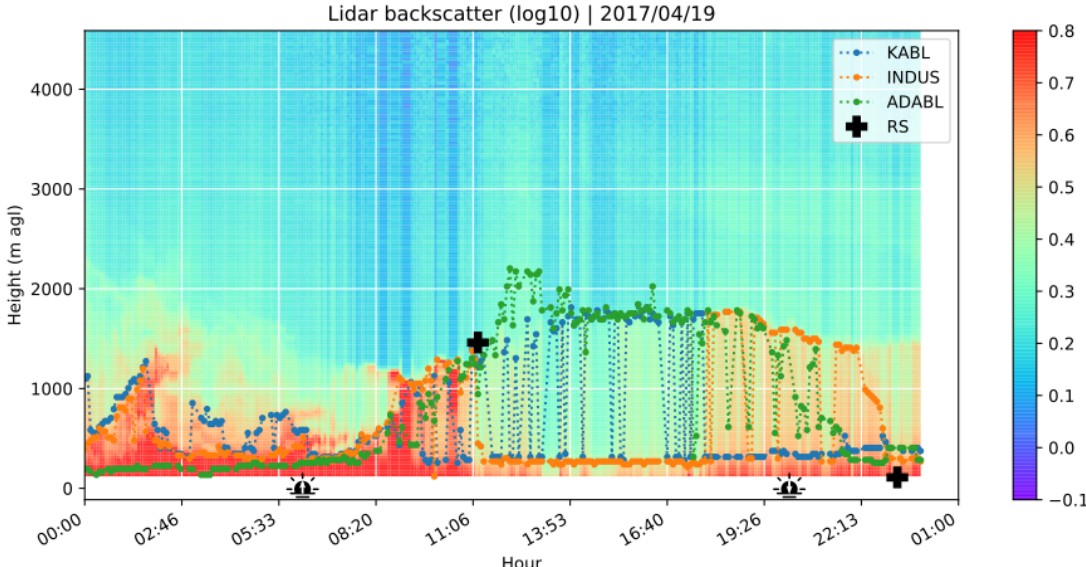

**Figure 11.** Case study: BLH estimates using different methods on April 19, 2017, at the Trappes site. Superimposed on the lidar range-corrected signal are BLH estimates from KABL (blue dotted line), the manufacturer's algorithm (denoted "INDUS", orange dotted line), and ADABL (green dotted line). The two crosses at 11:15 UTC and 23:15 UTC indicate the radiosonde estimates for that day and the two icons at 5:50 UTC and 19:49 UTC represent the time of sunrise and sunset, respectively.

However, KABL includes more irrelevant estimates (selecting remnants of the surface layer) than the other methods and ADABL gives an estimate that is too high for no apparent reason around 12:00 UTC. During the day, ADABL sticks to the top of the boundary layer, the manufacturer's algorithm sticks to the surface layer (which is very visible), and KABL oscillates between the two. The evening transition is blurry; the signal from the surface layer slowly increases, as the mixed layer decays into a residual layer. KABL locates this transition very early (around 17:00 UTC), when it stops oscillating and sticks to the 425 surface layer. ADABL makes the transition more smoothly, from 19:00 UTC to 22:00 UTC. The manufacturer's algorithm is the last to make the transition, at around 23:00 UTC, and the transition then occurs very sharply. We can conclude from this case study that none of the algorithms perfectly capture the boundary layer. Some of the limitations are physical, e.g., the evening transition is ill defined, resulting in disagreement between the algorithms. BLH-RS at 23:15 UTC is close to the lower boundary of the lidar range. This highlights the fact that BLH below 120 m are not rare and will not be detected by lidar if 430 BLH is in the lidar blind zone. Some of the other limitations are algorithmic; KABL has an unfortunate tendency to oscillate between several candidates for the top of the boundary layer (surface layer or clouds), and ADABL too closely reproduces the features of the days it has been trained on (e.g., night estimates and morning transitions).

## 5    Discussion and prospects

### 5.1    Algorithm maturity

Both algorithms examined here are not yet mature when applied in this context. K-means algorithms have already been used to detect BLHs in previous studies (Toledo et al., 2014; Caicedo et al., 2017; Toledo et al., 2017; Rieutord et al., 2014); therefore, it is a more mature method. This is visible in this paper via the level of investigation, which was much higher for KABL than for ADABL. Concerning boosting, this the first time, to our knowledge, that such an algorithm has been tested on this type of problem; therefore, ADABL is a completely new algorithm. Yet, it outperforms KABL and competes favorably with the

manufacturer's algorithm despite raising training issues.

### 5.2    Time and altitude continuity

The oscillations observed in Figure 11 are unrealistic and need to be avoided. They occur with KABL because clusters do not always have vertical persistence, as shown in Figure 12. One can see the cluster labels on a time–altitude grid for the same day as in Figure 11. When the initialization is random (Figure 12a, **init** = 'random', default settings in K-means), the labels

are also random. Only the transition of the labels on a profile is important. When the initialization is given (Figure 12b, **init** = 'given', retained settings in KABL), the labels can be identified. The blue cluster starts from very high attenuated backscatter coefficient (it detects clouds and the shallow morning BL); the red cluster starts from high attenuated backscatter coefficient (it detects the mixed layer or residual layer); and the green cluster starts from low attenuated backscatter coefficient (it detects the free atmosphere). Oscillations occur when some points are identified as free atmosphere in the middle of the boundary layer.

In the case study presented here (Figures 11 and 12b), this happens in the afternoon, when the blue cluster (starting from very high attenuated backscatter coefficient) gathers a few points near the first measurements and the overhead artifacts because very high attenuated backscatter coefficient is irrelevant under these conditions.

Several prospects exist to enforce vertical persistence of the clusters in KABL, we list here four examples. First, the problem identified in Figure 12b could be solved by choosing the number of clusters automatically. However, this option was tested

(case where **n_clusters** = 'auto' in Sect. 3.3) and it did not solve this issue. The sensitivity analysis showed that **n_clusters** = 'auto' led to higher discrepancies with respect to BLH-RS. In fact, this setting usually gives low estimates of BLH because the automatically chosen number of clusters is usually high. A more advanced strategy to automatically choose the number of clusters (e.g., Tibshirani et al., 2001) might get around this issue. The vertical persistence of clusters can be enforced by adding altitude to the KABL predictors. This can be done thanks to a post-processing, for example, with a moving average

or by imposing a maximum BLH growth rate (Poltera et al., 2017). The distance used in K-means could also be modified to incorporate these constraints, for example, by adding penalty terms.

In ADABL, time and altitude continuity are ensured because they are within the predictors. However, ADABL yields BLH estimates that are too similar to the BLHs in the training set. Removing time and/or altitude from the predictors should be considered to force the algorithm to rely more on the measurements. Further, the sensitivity analysis presented here for KABL

needs to be performed for ADABL.

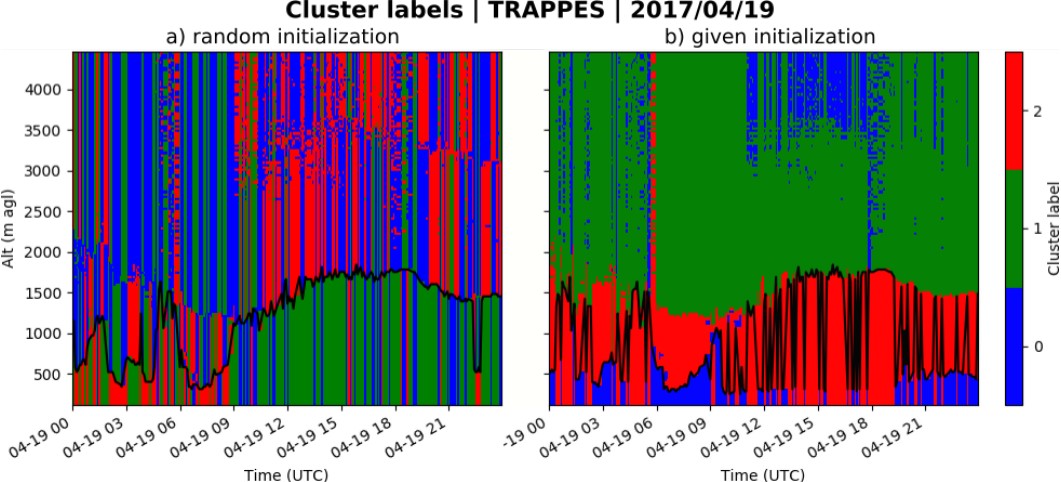

**Figure 12.** Cluster labels for KABL output on a time–altitude grid for April 19, 2017, at Trappes with KABL BLH (black line) estimated using (a) the centroids initialized at random and (b) the centroids initialized at given values.

## 5.3 Real-time estimation

Even though it was not necessary for this study, all of the algorithms studied here can be used in real time. As soon as a lidar profile is available, BLH estimations can be performed instantaneously[4]. However, KABL suffers from undesirable oscillations from one profile to the next. A method to filter these oscillations is needed but would disable the "real-time" feature of the algorithm. In addition, the hour of the day needs to be explicitly passed in a periodic function. This has not been done here because we worked only on 24-hour time periods.

## 5.4 Quality of the evaluation

Even though we made an effort to sort the meteorological conditions using ancillary data, the two-year comparison still mixes heterogeneous conditions. In addition, the results are clearly different at the sites studied here, emphasizing the importance of local conditions. A more precise casting of the meteorological conditions with atmospheric stability indices or large-scale insights would lead to a better understanding of the strengths and weaknesses of the algorithms. The importance of the sites needs to be investigated by extending the study to a larger number of sites with different environments. A more careful examination of cloudy days also needs to be performed. Cases where the cloud bases are below 3 km were filtered from our study. However, cases where clouds reside inside the inversion should be detected by KABL as an extra cluster; further studies are required to confirm this behavior. In addition, ADABL was not specifically trained to deal with cloudy situations. Further studies to determine how ADABL behaves without training and how it could be appropriately trained are required.

---

[4]Both KABL and ADABL need less than 1 s to run a single profile.

## 5.5 Quality of the reference

Radiosonde profiles are usually regarded as the best reference for BLH. However, the derivation of BLH from such measurements is contentious because several methods exist and some strongly disagree. Moreover, RS measurements cannot be used to assess the full diurnal cycle of BLH. This is a clear limitation of this study because the RS measurements cannot determine if the difference between the diurnal cycle of ADABL and those of the other methods represents an improvement. Therefore, a very interesting project would be to use a dedicated field experiment with high-frequency tethersonde or other continuously running instruments as a reference. For example, microwave radiometers are good candidates because they provide information that is not based on aerosols and the derivation of BLH from these instruments are routine (Cimini et al., 2013).

## 5.6 ADABL: Training

ADABL already shows good performance when trained on only two days. Most of its bad estimates result from the short length of its training period. Therefore, a short-term project would be to label more days with various meteorological conditions. However, the dependence of ADABL on training makes it sensitive to instrumentation settings and calibrations. Even though the effect of a calibration or the evolution of an instrumental device has not been studied, it is likely that training needs to be repeated after each calibration or change in the instrumental device. Therefore, two strategies are possible for training ADABL: remove the influence of calibration prior to training (this would require knowing the instrumental constants for all of the devices) or train it to deal with differences (this would require including as many different devices as possible in the training set, which would then become very large). In any case, the main limitation will be the need to label the entire training dataset (a priori by human experts).

## 5.7 KABL: Training-less

KABL appeared to perform the least well in this study; however, there are interesting prospects to improve its performance. KABL does not require any training; therefore, it is less dependent on instrumentation settings and calibrations. Because it is not strongly dependent on the instrumental devices, it can be used on backscatter profiles made by other instruments (e.g., ceilometers). Moreover, other profiles besides the backscatter intensity can be added as additional predictors for unsupervised learning after normalization. Therefore, the concept of KABL can be advanced further to create synergy between multiple remote sensing instruments. Microwave radiometers are good candidates because they have comparable time resolution to lidar and provide independent information concerning the thermal stratification of the boundary layer. Cloud radars also have comparable time resolution to lidar and provide additional independent information.

## 5.8 Quality flags

Currently, no quality flags for the estimation are provided. One approach would be to use the internal scores (i.e., silhouette, Davies–Bouldin, and Calinski–Harabasz defined in Sect. 3.4) as quality flags; however, further study is required to determine whether these metrics can serve as reliable quality flags.

## 6 Conclusions

This paper described two algorithms based on machine learning to estimate the boundary layer height from aerosol lidar measurements. The first, KABL, is based on the K-means algorithm. The second, ADABL, is based on the AdaBoost algorithm. Both algorithms take the same input file, one day of data generated by the *raw2l1* routine, and produce similar output, a BLH time series for the input day. KABL is a non-supervised algorithm that looks for a natural separation in the backscatter signals between the boundary layer and the free atmosphere. ADABL is a supervised algorithm that fits a large number of decision trees in a labeled dataset and aggregates them in an intelligent manner to provide a good prediction. KABL, ADABL, and the lidar manufacturer's algorithm were tested on a two-year dataset taken from the Météo-France operational lidar network. The Trappes and Brest sites were chosen because of their different climates and the availability of regular RS measurements, which were used as a reference.

A large discrepancy in RMSE and the correlation with the radiosondes was observed between the two sites. At the Trappes site, KABL and ADABL outperformed the manufacturer's algorithm while the opposite occurred at the Brest site. At both sites, ADABL performed better than KABL (higher correlation and lower error) and the manufacturer's algorithm performed well. By analyzing the seasonal and diurnal cycles, we determined that the KABL and manufacturer's estimates have similar behavior; however, the KABL estimates are always higher by approximately 200 m. ADABL generates the most pronounced diurnal cycle, with a pattern that is very similar to the expected diurnal cycle; however, its results depend greatly on the days it has been trained on. In particular, the sunset and sunrise times of these days over-influenced the ADABL estimate. In the case study, we saw that both algorithms perform well overall; however, we identified several algorithmic limitations, e.g., KABL tended to oscillate between several candidates for the top of the boundary layer (surface layers or clouds) and ADABL was overly constrained by the days it was trained on (e.g., the night estimate and morning transition). In summary, ADABL is promising but has training issues that need to be resolved, KABL has a lower performance but is much more versatile, and the manufacturer's algorithm using a wavelet covariance transform performs well with little tuning but is not open source. A wide range of future developments is available for ADABL and KABL, the most immediate being that the training set of ADABL can be enhanced, time and altitude continuity can be enforced in the KABL estimation, and both can be compared to high temporal resolution RS measurements.

*Code availability.* The KABL source code is available to and usable by all users, including commercial users. The code is freely available under an open-source license at the following link: https://github.com/ThomasRieutord/kabl. It is made in Python 3.7 with regular statistics and machine-learning packages, namely, Scikit-learn 0.20 (Pedregosa et al., 2011) and SALib 1.3.7 (Herman and Usher, 2017), which are open source and available under free licenses. The repository contains all the necessary features to run the code on *raw2l1* outputs. Several days of data are also provided as examples.

*Author contributions.* Tiago Machado implemented an initial version of the KABL code and performed the first comparisons to the RS data. Sylvain Aubert extracted and processed all the data (lidar, radiosonde, and ancillary), made one of the hand-labeled BLHs, and participated actively in the writing of the manuscript. Thomas Rieutord implemented the current versions of KABL and ADABL, made one of the hand-labeled BLHs, produced the figures, and actively participated in the writing of the manuscript.

*Competing interests.* The authors declare that they have no conflicts of interest.

*Acknowledgements.* The authors would like to thank Alexandre Paci, Alain Dabas, and Olivier Traullé for their helpful reading and comments. We would like to thank Marc-Antoine Drouin, from the Site Instrumental de Recherche par Télédétection Atmosphérique, for providing us with the link to *raw2l1* and all the Météo-France agents who install and maintain the lidar network. We thank Martha Evonuk from Evonuk Scientific Editing (http://evonukscientificediting.com) for editing a draft of this manuscript.

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
