# Peer review of "Deriving boundary layer height from aerosol lidar using machine learning: KABL and ADABL algorithms"

_Atmospheric Measurement Techniques, 2020_

## Referee Comment (RC1) · Anonymous Referee #1 · 29 Apr 2020

This article relates a study dedicated to evaluate the performance of two algorithms based on machine learning for retrieving the ABL height from lidar observations. To this end, the study makes use of a 2-year series of observations acquired by two lidars operating at two different locations. From these observations the authors retrieve the ABL height using KABL and ADABL algorithms and compare them with those obtained from radiosondes measurements (taken as reference). Based on this comparison the authors argue that better results are obtained with ADABL approach, although indicate that KABL can be easily adapted for 'other instrumental device'. The manuscript is not concisely written, and lacks scientific novelty and significance, or at least in the way it is written. The description of the methods is rather confusing and the authors make use of a large number of brief sections that are not well presented and discussed (e.g.

[Figure]

section 5). The authors also fail to demonstrate their main conclusions about the two algorithms. Specific comments and suggestions are given below:

- The abstract lacks of motivation. Currently, there are a number of different methods to retrieve the ABL from lidar observations, whose results have been widely tested. Why should we use KABL and ADAABL algorithms?

-The abstract states that "ADABL algorithm is performing better than KABL...". However, the authors do not indicate what they mean with 'performing better', or based on which results. In addition, the comparison uses radiosondes that always launched at the same time. How might this affect the study findings?

-Radiosonde data section, last sentence: 'After testing some of these methods on our dataset, we chose to derive boundary layer height with parcel method for the 11:15 sounding and bulk Richardson number for the 23:15 one.' Since the ABL height retrieved from radiosonde data is taken as reference in this study, the authors must explain why the parcel and Richardson methods were chosen. Also, they must explain how these different methods were tested.

-Section 3.1.2 and figure 4: The authors state "For few days where the boundary layer is easily visible for a human expert, the boundary layer top is drawn by hand: all points below this...". I do not believe this is a criterion to estimate the ABL height. I suggest to use some of the other methods that have been previously tested and used in the literature.

-Figure 9 shows the RMSE and correlation obtained from their comparison. However, nothing is said regarding RMSE. How did they calculate it? How is it defined? What are the errors of the radiosonde and lidar retrievals? Does the correlation depend on the ABL height? Also, it is said that a number of cases were not included in the analysis as a result of the meteorological conditions. How many cases? Are the retrievals affected by the meteorological conditions, why?

In summary, the overall comparison is incomplete and does not convincingly demonstrate their main conclusions.

---

## Referee Comment (RC2) · Anton Sokolov (Referee) · 5 May 2020

In the paper "Mixing height derivation from aerosol lidar using machine learning: KABL and ADABL algorithms" two machine learning algorithms presented for the definition of the atmospheric boundary layer height, which is a principal parameter for the atmospheric modelling and air pollution dispersion. The detection of ABL is not a straightforward task, and even though the precision of proposed algorithms is mediocre, I think that the article contains results that could be interesting to the scientific community and correspond well to the scope of the Atmospheric Measurement Techniques journal. An important positive point is that developed programs are available on the internet at the GitHub site under a fully open access option.

[Figure]

Nevertheless, there are a few issues on training, validation and testing, that I mentioned below in the General comments section. In my opinion, the quality of English should also be improved, and the paper would benefit from proofreading by a native speaker. Some theoretical explanations and expressions are often not accurate. There are also problems with text structure: some variables appear in the text before explanations in later paragraphs.

General (Major) Comments

1. When training the supervised ML algorithm, the estimation of the accuracy of ADABL on the validation ensemble (by the cross-validation technique) is presented in line 171: 99.5%. I think it is important to present also the accuracy on some testing ensemble, at least on the case study of April 19, 2017. It will justify the generalization ability of the applied AdaBoost algorithm showing the algorithm performance in an independent dataset. In my opinion, the training dataset could be insufficient.

2. Another reason why the accuracy of ADABL on validation ensemble is unrealistically high is the application of the random cross-validation split for time-correlated data (line 171). Using random selection from correlated datasets can lead to loss of generalization ability of the algorithm. The proposed ML method should be applied to new (independent) measurements. It means that AdaBoost should be trained-validated on uncorrelated parts of the dataset. If the data points were selected randomly from the whole dataset by the cross-validation procedure, it is highly probable that the similar neighbouring time points would be placed in both training and validation ensembles, which gives unrealistically good accuracy estimate on training-calibration datasets, but the worse result on another independent (test) dataset. I suggest the application of the block cross-validation.

3. If I understand right, the final configuration of the unsupervised ML algorithm (KABL) produces the classification using just one parameter - RCS0. In this case, the phrase in the conclusion (line 435) is misleading – "Both take the same input: one day of data

generated by raw2l1 routine; . . ."

4. Line 272: "number of invalid values (NaN or Inf) are recorded." - Please explain why algorithms return these kinds of values. Another question is how algorithms deal with undefined values in Lidar measurements. Specific (Minor) comments

5. Line 15: ". . . boundary layer height (BLH). . ." – please give somewhere a definition of the BHL.

6. Line 105: ". . . we chose to derive boundary layer height with parcel method for the 11:15 sounding and bulk Richardson number for the 23:15 one." Please justify why two different methods were used for morning and evening radiosounding.

7. Line 113: Does false positives on cloud detection perturb a BLH detection? Please explain.

8. Line 113: In the following text, some basic ML concepts are introduced for readers, who are not familiar with the scope of ML. In this case, the "false positives" should also be explained or referenced.

9. Line 127: As the number of seconds, since midnight is a periodical function, the 'classical' distance could not take it in consideration correctly this variable. It means that the classical distance between one 00:01 and 23:59 will be nearly 24 hours. Please make sure, that ADABL algorithm works as expected in this case.

10. Line 142: I do not see any subsampling in figure 3. Is it a five-forks weak learner creation part? Please specify.

11. Line 142: How these shallow decision trees are fitted? I have never heard about resampling in the classical AdaBoost. Is it Bagging? Please give a reference or explain the algorithm in detail.

12. Line 143: ". . .the error of the classifier is the number of misclassified points." - I am not sure that error is defined like that. Please explain or give a reference.

13. Line 146: The explanation is not sufficient. I propose to present here a reference to any popular textbook on AdaBoost or carefully introduce the algorithm. For example, in the expression, the performance was not introduced, the upper limit in the sum should be capitalized, etc...

14. Line 170: "trade-off between accuracy and computing time" - I do not think that the limiting factor for this problem is the computing time. Normally this kind of problem could be sufficiently well resolved by parallel computing.

15. Line 169: "RCSco, RCScr" – please make sure that these names for copolarized and crosspolarized range-corrected backscatter signals persist in the following text (notably in tables 2 and 3).

16. Line 175: "It is possible to quantify the relative importance of the predictors (Breiman et al., 1984; Hastie et al., 2009). After the training, the time accounts for 30.3%, RCSco for 28.4%, RCScr for 26.5% and the altitude for 14.8%." – I have not found this information; could you please specify the corresponding page numbers?

17. Line 184: "distances from all points to all centroids" – Are these the Euclidian distances?

18. Line 196: "If we assume all Gaussian have the same fixed variance and that this variance tends to zero, EM and K-means algorithms are the same." – Could you provide a reference or explain the statement?

19. Line 203: "Then the data they contain are normalized. . ." – if time and height are used in the KABL algorithm, are these variables also normalized?

20. Line 205: What values are included in predictors? If X matrix contains only signals RCSco and RCScr, it should be stated somewhere.

21. Line 209: "Finally, we look for the first change in clusters attribution, starting from the ground. This gives us the BLH for this profile." I am not sure that this algorithm is optimal, as it could lead to oscillations of BLH. To understand how it could be improved

I suggest presenting and analyzing the altitude-time plot with pixels representing the results of the classification (like Fig 11 but with classes). Probably it is better not to take the first change, but a height above which the class is not changing, e.g. for three levels. Alternatively, a value of height could be selected that persist in time. These kinds of parameters could be optimally selected by the scores optimization. Another option is to modify the 'distance' definition.

22. Line 213: "The parameters of this computer code..." these parameters should be introduced at the beginning of the section 3.3, before they are referenced.

23. Line 298: "... figure 8 the distribution (violin plots) of the relevant output conditionally to the parameter value." – I suggest adding here a reference on the construction of this kind of plot.

24. Line 304: "Parameters values are chosen to give the most optimal value for the metrics they have influence on." - The selection of locally optimal combination of parameters does not provide the globally optimal solution. How can you be sure that this combination gives the best precision?

25. Line 316: "As the average gap E1 and the RMSE E2 are very similar..." – I suggest excluding the average gap E1 from the article for the sake of simplification.

26. Line 326: "Nighttime (launch of 23:15 UTC)" - If nighttime radiosounding was not used, why to present this dataset in "2.2 Radiosonde data"? Probably it was used in supervised ML? Please specify.

27. Figure 9: Adding the confidence intervals for RMSE and Correlation in Fig. 9 could be quite useful.

28. Line 335 and Line 438: I think it would be advantages to understand how works the lidar manufacturer's software and to give some interpretations.

29. Line 399: "A method to filter these oscillations will be needed, but it can also divest the "real-time" property." – Instead of filtering, the criteria of the lowest transition of the

class for KABL could be somehow modified, as I proposed in my comment for line 209. The filtering could be of the "real-time" if it is done relatively the past classifications.

30. Line 422: "5.6 KABL is "trainingless"" – I suggest that KABL could be used also by an expert to simplify the learning stage of supervised ML.

31. Line 417: "...strategies ... for the training of ADABL..." – To decrease the sensitivity to "idealized" diurnal cycle of the BLH, I suggest trying to exclude the time predictor in ADABL.

Technical corrections

32. Line 13: "Atmospheric boundary layer concentrates many scientific challenges (small scale flows, turbulence...) and with high impacts due to its position of the interface between ground and atmosphere." - awkward English, please correct.

33. Line 27: "(clouds, residual layers..)" -> "(clouds, residual layers...)".

34. Line 88: "SIRTA" - Please decrypt the abbreviation.

35. Line 92: "at 11:15 AM and PM" – Please utilize the same notation for the time here and further. I suggest the UTC format.

36. Line 94: Please explain what the theta is. Is it the potential temperature?

37. Line 127: I suggest inserting a comma after "height above ground".

38. Line 129: "[[1,N]]"- What does double brackets means? Please explain.

39. Line 145: "tree" -> "trees".

40. Line 146: "m=200" -> "M=200".

41. Line 154, 157, fig. 4: "top"->"left", "bottom"->"right".

42. Line 208: Init parameter is not defined.

43. Line 208: "specified in algo" -> "specified by algo parameter"

[Figure]

44. Line 270: "(0.20)" Is it the software version? Please specify.

45. Figure 10: Please introduce the INDUS abbreviation.

---

## Author Comment (AC1) · 30 Jun 2020

The referee points out relevant critics that have passed through the internal review among authors before submission. The authors acknowledge that greater effort must be made in the redaction of the study. They would like to defend the scientific novelty of their work in the joint point-by-point response.

Please also note the supplement to this comment:
https://www.atmos-meas-tech-discuss.net/amt-2020-78/amt-2020-78-AC1-supplement.pdf
* * *
[Figure]

**Supplement:**

**Q1 "The abstract lacks of motivation. Currently, there are a number of different methods to retrieve the ABL from lidar observations, whose results have been widely tested. Why should we use KABL and ADAABL algorithms?"**

KABL and ADABL are open-source, usable by anyone, and not strongly tied to a particular instrument. Several methods exist to derive ABL from lidar observations, but it is also acknowledged that none of them are fully satisfying and the research on this topic is still active.

KABL is the reproduction of an existing method (scientifically interesting to reproduce previous results) with open-source libraries (technically interesting to give access to the source code and allow its reuse). ADABL is a new algorithm (also open-source) that enables the reproduction of human expertise that we believe is valuable for deriving BLH. We develop this last point in our answer to the question 4, later in this document.

**Q2 "The abstract states that "ADABL algorithm is performing better than KABL...". However, the authors do not indicate what they mean with 'performing better', or based on which results. In addition, the comparison uses radiosondes that always launched at the same time. How might this affect the study findings?"**

The statement "ADABL algorithm is performing better than KABL. . ." refers to correlation and RMSE with RS, as both are better for ADABL.

The comparison with RS launched only twice a day is indeed a strong limitation of the evaluation. First, it reduces drastically the amount of KABL and ADABL estimations subject to evaluation (only 0.7 % are evaluated). Second, the RS are launched at 11:15 UTC and 23:15 UTC which are respectively at the end of the morning transition and at night. Such periods are known to be problematic for an accurate ABL estimation (for all methods). Third, the evolution of ABL throughout the day, which is important to assess the estimation quality, cannot be evaluated.

Q3 "Radiosonde data section, last sentence: 'After testing some of these methods on our dataset, we chose to derive boundary layer height with parcel method for the 11:15 sounding and bulk Richardson number for the 23:15 one.' Since the ABL height retrieved from radiosonde data is taken as reference in this study, the authors must explain why the parcel and Richardson methods were chosen. Also, they must explain how these different methods were tested."

The chosen rule "parcel method for 11:15 and bulk Richarson number at 23:15" follows the recommendations of Seibert et al. (2000), figure 10, assuming that morning launch is in unstable atmosphere and evening launch is in stable atmosphere.

However, our experience showed that estimation of BLH with the RS is not straightforward, even though they are considered as the reference. Several methods were implemented and applied on the 2-year data, but they hardly match, as it is shown in this figure :

---

## Author Comment (AC2) · 30 Jun 2020

The authors would like to thank Dr Sokolov for his thorough review which is noticeably beneficial for the manuscript. All raised concerns are addressed in the joint point-by-point response. As soon as the scientifically-motivated modifications are done, the manuscript will get checked by a native English-speaker.

Please also note the supplement to this comment:
https://www.atmos-meas-tech-discuss.net/amt-2020-78/amt-2020-78-AC2-supplement.pdf

[Figure]

**Supplement:**

**General (Major) Comments**

**1. When training the supervised ML algorithm, the estimation of the accuracy of ADABL on the validation ensemble (by the cross-validation technique) is presented in line 171: 99.5%. I think it is important to present also the accuracy on some testing ensemble, at least on the case study of April 19, 2017. It will justify the generalization ability of the applied AdaBoost algorithm showing the algorithm performance in an independent dataset. In my opinion, the training dataset could be insufficient.**

We acknowledge that the training dataset is insufficient, as it is commented in the results section. The point of this study is to show that, despite few training, ADABL has reasonably good results. As data labeling and algorithm training are time-consuming tasks, we would like to demonstrate first doing such tasks is of interest.
Time-consumption was also the reason why the case study of April 19 was not labeled, but the referee makes a very good point arguing that this would enable us to make a more accurate evaluation of ADABL (and KABL as well) on an brand new day. It will be added in the short term prospects.

In order to give a lower bound for the accuracy, we trained ADABL on one day and used the second day as validation set. The resulting accuracy is 87.6%. It is noticeably lower than the previous figure, but it suffers from a half-less rich training set.
In any case, the accuracy value is not meant to be a finding of this study. We extent this opinion in the next comment which is closely related.

**2. Another reason why the accuracy of ADABL on validation ensemble is unrealistically high is the application of the random cross-validation split for time-correlated data (line 171). Using random selection from correlated datasets can lead to loss of generalization ability of the algorithm. The proposed ML method should be applied to new (independent) measurements. It means that AdaBoost should be trained-validated on uncorrelated parts of the dataset. If the data points were selected randomly from the whole dataset by the cross-validation procedure, it is highly probable that the similar neighbouring time points would be placed in both training and validation ensembles, which gives unrealistically good accuracy estimate on training-calibration datasets, but the worse result on another independent (test) dataset. I suggest the application of the block cross-validation.**

We agree with the reviewer that the accuracy as it is estimated is too optimistic. Our effort were not directed towards having the best estimation of accuracy because it was only used to discriminate different supervised algorithms. The result of the comparison is not shown in the manuscript but can be found here: https://presentations.copernicus.org/EGU2020/EGU2020-19807_presentation.pdf
To meet with the referee's concern, the same study has been repeated with a block cross-validation. It was performed with group K-fold where groups were 4 hours chunks. The code to generate the figure is online on the Github repository (examples/perform_block_cv.py). The results are in the following figure. We can conclude that AdaBoost is still the most accurate among the one tested and its accuracy is rather 0.96 than

0.99.

[Figure]

**Performance/speed comparison of estimators**

To conclude, the accuracy value will be changed in the manuscript. The value of 0.96, obtained with block cross-validation is the most relevant in our opinion because it is more realistic than 0.99 and it was compared to the accuracy obtained by other algorithms with the same calculation. The lack of validation set will be highlighted as a limitation of the study.

**3. If I understand right, the final configuration of the unsupervised ML algorithm (KABL) produces the classification using just one parameter - RCS0. In this case, the phrase in the conclusion (line 435) is misleading – "Both take the same input: one day of data generated by raw2l1 routine; . . ."**

The final configuration of KABL does use only one parameter. But both algorithms still take the same input: a daily file generated by raw2l1 that contains all information needed for both algorithms, each algorithms extract only what it needs.

**4. Line 272: "number of invalid values (NaN or Inf) are recorded." - Please explain why algorithms return these kinds of values. Another question is how algorithms deal with undefined values in Lidar measurements.**

Algorithms return NaNs when all the points of the profile are assigned to the same cluster. For ADABL, it happens when the profile is very different than the ones in the training set (not that rare). For KABL, it happens when we specified the initial centroids (it the case in the retained configuration) and only one of these points gather a cluster around it (very rare). When lidar has few undefined values in the profile, they are

just ignored and the estimation is made on the available points.

**Specific (Minor) comments**

**5. Line 15: ". . . boundary layer height (BLH). . ." – please give somewhere a definition of the BHL.**

BHL is probably a typo error for BLH, as a research for "BHL" inside the manuscript returned no results. The BLH definition we used here is in the next sentence (lines 15-16) as "the depth of atmosphere where all pollutants emitted from the ground will remain".

**6. Line 105: ". . . we chose to derive boundary layer height with parcel method for the 11:15 sounding and bulk Richardson number for the 23:15 one." Please justify why two different methods were used for morning and evening radiosounding.**

We followed the recommendations in Seibert et al. (2000), figure 10, assuming that morning launch is in unstable atmosphere and evening launch is in stable atmosphere.

As the other referee had a major comment about the method used for RS, a more complete answer was given. It might be of interest here if any additional question arise.

**7. Line 113: Does false positives on cloud detection perturb a BLH detection? Please explain.**

Cloud screening with the collocated CL31 was only used to exclude cases for the comparison with the radiosoundings. Therefore, false positives in cloud detection would have for only effect to improperly reduce the comparison sample. The MiniMPL detection of clouds was found to reduce too much the comparison sample, while CL31 detection of clouds looked more reliable.

**8. Line 113: In the following text, some basic ML concepts are introduced for readers, who are not familiar with the scope of ML. In this case, the "false positives" should also be explained or referenced.**

The false positives refer here to the detection of clouds, not to the BLH detection.

To avoid confusion, the following sentences "Although the MiniMPL (...) to make some false positives." will be replaced by "Although the MiniMPL is perfectly capable of detecting clouds, we relied more on the cloud detection with the CL31, because MiniMPL's cloud detection was detecting cloud where there was not."

After the correction of the misleading use of words "false positives" and "algorithms" line 113, we do not think it is necessary to introduce false positive in section 3.

**9. Line 127: As the number of seconds, since midnight is a periodical function, the 'classical' distance could not take it in consideration correctly this variable. It means**

**that the classical distance between one 00:01 and 23:59 will be nearly 24 hours. Please make sure, that ADABL algorithm works as expected in this case.**

This remark is very important for the next stages of the algorithms development. However, at the moment, both algorithms have been used only on 24 hours chunks, so that the periodicity was not an issue.

**10. Line 142: I do not see any subsampling in figure 3. Is it a five-forks weak learner creation part? Please specify.**

It is correct to see no subsampling in the figure 3: AdaBoost do not perform subsampling, as the referee points out in the next comment.

Figure 3 is only meant to illustrate the boosting algorithm (as figure 2 is only meant to illustrate decision tree). They do not reflect the actual settings of the algorithms, which would needlessly blur the pedagogical effort. The decision trees used as weak learner are five-forks trees or less.

**11. Line 142: How these shallow decision trees are fitted? I have never heard about resampling in the classical AdaBoost. Is it Bagging? Please give a reference or explain the algorithm in detail.**

The referee raises an error here: indeed the weak learners are not trained on a subsample but on the whole dataset. Although, the weight put on each sample changes from one weak learner to another. Weak learners are trained with CART algorithm. A very good description of the algorithm can be found in Hastie et al. (2009) page 307.

**12. Line 143: ". . .the error of the classifier is the number of misclassified points." - I am not sure that error is defined like that. Please explain or give a reference.**

We acknowledge this sentence is incorrect: the error is the weighted average of misclassified points. For more details, the formula is given in Hastie et al. (2009) page 339, algorithm 10.1. The actual implementation can be checked in the Scikit-learn source code: sklearn.ensemble._weight_boosting.py: line 528.

**13. Line 146: The explanation is not sufficient. I propose to present here a reference to any popular textbook on AdaBoost or carefully introduce the algorithm. For example, in the expression, the performance was not introduced, the upper limit in the sum should be capitalized, etc.**

The reference to Hastie et al. (2009), which is a popular textbook explaining the algorithm with many details and well written, was made at the section 3.1. It is completed by the reference to Freund & Schapire (1997), which is the original publication of AdaBoost and contains many theoretical results. The authors will complete theses references with Schapire, R. E. (2013). Explaining adaboost. In *Empirical inference* (pp. 37-52). Springer, Berlin, Heidelberg.

This paragraph was kept short to avoid overwhelming the reader with technical details. We will reformulate

in order to keep the main idea and refer to the literature for the details.

**14. Line 170: "trade-off between accuracy and computing time" - I do not think that the limiting factor for this problem is the computing time. Normally this kind of problem could be sufficiently well resolved by parallel computing.**

It is true that computing time is not critical here because it is always low, even without parallel computing. However, we do not want a classifier needlessly complex. For example, fully-grow trees would be long to train and test for very little extra-performance.

As computing time is not a major factor, we will change the sentence line 170 by "It was chosen because more complex classifier do not show greater performance."

**15. Line 169: "RCSco, RCScr" – please make sure that these names for copolarized and crosspolarized range-corrected backscatter signals persist in the following text (notably in tables 2 and 3).**

RCSco and RCScr are named respectively RCS1 and RCS2 in the source code. We chose to use RCSco and RCScr in the text because it is less ambiguous and to use RCS1 and RCS2 in the table so that readers going through the code would not be confused.
As this choice seems to be confusing for reader, we will keep only RCSco/cr in the paper and make the correspondence within the code

**16. Line 175: "It is possible to quantify the relative importance of the predictors (Breiman et al., 1984; Hastie et al., 2009). After the training, the time accounts for 30.3%, RCSco for 28.4%, RCScr for 26.5% and the altitude for 14.8%." – I have not found this information; could you please specify the corresponding page numbers?**

In Hastie et al. (2009), it can be found page 368, equation 10.43.
In Breiman et al. (1984), it can be found page 147, definition 5.9.

**17. Line 184: "distances from all points to all centroids" – Are these the Euclidian distances?**

Yes, K-means usually implies Euclidean distance. Although it is technically possible to use any distance, most of implementations do not provide this option.

**18. Line 196: "If we assume all Gaussian have the same fixed variance and that this variance tends to zero, EM and K-means algorithms are the same." – Could you provide a reference or explain the statement?**

In Hastie et al. (2009), exercice 14.2, page 580.
Let's consider a sample generated by a mixture of two Gaussian with the same fixed variance. When the variance tends to zero, the "expectation" step is the same as the attribution to the closest centroid, and the "maximisation" step is the same as updating the centroids.

**19. Line 203: "Then the data they contain are normalized. . ." – if time and height are used in the KABL algorithm, are these variables also normalized?**

Time and height are not used in KABL algorithm. However, all the data are normalized in order to avoid any unit comparison problem. The normalization consists in removing the mean and divide by the standard deviation. It is done in kabl.core.py: line 152 (release 1.0.0. Current commit: line 167).

**20. Line 205: What values are included in predictors? If X matrix contains only signals RCSco and RCScr, it should be stated somewhere.**

Predictors can be either RCSco all day, either RCSco and RCScr all day or RCSco for daytime and both RCSco and RCScr for nighttime, as precised in Table 2.
This question arises because of the ill-suited organisation of the text, as pointed out by comment 22, but this will be corrected.

**21. Line 209: "Finally, we look for the first change in clusters attribution, starting from the ground. This gives us the BLH for this profile." I am not sure that this algorithm is optimal, as it could lead to oscillations of BLH. To understand how it could be improved I suggest presenting and analyzing the altitude-time plot with pixels representing the results of the classification (like Fig 11 but with classes). Probably it is better not to take the first change, but a height above which the class is not changing, e.g. for three levels. Alternatively, a value of height could be selected that persist in time. These kinds of parameters could be optimally selected by the scores optimization. Another option is to modify the 'distance' definition.**

We do observe some oscillations as the referee describes (e.g. see figure 11). The proposition made by the referee, to enforce vertical persistence of the clusters, is very relevant and will be added as a prospect. The time continuity is also very relevant, and existing methods already use such criteria, thus are a source of inspiration to implement it here. The distance definition is probably the "smartest" way, as it can help us learn about what really matters to distinguish boundary layer from the rest, but it is also the less intuitive. The optimization of parameters was done here thanks to global sensitivity analysis, but it would greatly benefit from being repeated after such new features are added.

The following figure shows the altitude-time plot of the classes attributed by KABL, with random initialization (K-means "vanilla"). It makes very visible the random attribution of the classes numbers in unsupervised classification: only borders matter.

[Figure]

A way to avoid such random attribution is to specify the initial centroids: the resulting plot is next. Initial centroids were put at typical backscatter values (modes of the histogram). The blue cluster has a very high backscatter (it detects cloud and shallow morning BL), the red cluster has high backscatter (it detects mixed layer or residual layer), the green cluster has low backscatter (it detects free atmosphere). We can see patches of blue in the free atmosphere: they are not realistic. They occur in profiles were there is no strong backscatter corresponding to this cluster. As we still ask for 3 clusters, the blue cluster sticks to noisy points.

The BL top defined as the height where cluster stop changing, as suggested by the referee, would be affected by such noisy points.

[Figure]

**22. Line 213: "The parameters of this computer code. . ." these parameters should be introduced at the beginning of the section 3.3, before they are referenced.**

The organisation of this paragraph will be changed accordingly.

**23. Line 298: ". . . figure 8 the distribution (violin plots) of the relevant output conditionally to the parameter value." – I suggest adding here a reference on the construction of this kind of plot.**

Here is a reference describing the plot and its use:
Hintze, J. L., & Nelson, R. D. (1998). *Violin Plots: A Box Plot-Density Trace Synergism. The American Statistician, 52(2), 181–184.*

**24. Line 304: "Parameters values are chosen to give the most optimal value for the metrics they have influence on." - The selection of locally optimal combination of parameters does not provide the globally optimal solution. How can you be sure that this combination gives the best precision?**

This is a good remark: we cannot be sure. However, a sensitivity analysis on the 2-year long dataset would be computationally too expensive. The sensitivity analysis on a single day has the advantage to be more thorough than anything we could have done on the whole dataset. It helped us to selected few configurations to test on the whole dataset, including the configuration described in Table 3, which was elected because it had the best results. A sentence will be added to make clear that the sensitivity analysis on one day was used to select few configurations to be tried on two years.

**25. Line 316: "As the average gap E1 and the RMSE E2 are very similar. . ." – I suggest excluding the average gap E1 from the article for the sake of simplification.**

Yes, this is something we will do in the next version of the manuscript.

**26. Line 326: "Nighttime (launch of 23:15 UTC)" - If nighttime radiosounding was not used, why to present this dataset in "2.2 Radiosonde data"? Probably it was used in supervised ML? Please specify.**

Nighttime launches were not used in the supervised algorithm.
They were only used for the comparison to the diurnal BL cycles presented in Fig. 10.

**27. Figure 9: Adding the confidence intervals for RMSE and Correlation in Fig. 9 could be quite useful.**

Yes it would. We will add bootstrap estimations of confidence intervals in the next version of the manuscript.

**28. Line 335 and Line 438: I think it would be advantages to understand how works the lidar manufacturer's software and to give some interpretations.**

We asked the manufacturer for more details about their algorithm: a modified wavelet transform method described in Brooks, 2003 is used. In that regard, we will expand on the interpretations of the results for the revised manuscript.

**29. Line 399: "A method to filter these oscillations will be needed, but it can also divest the "real-time" property." – Instead of filtering, the criteria of the lowest transition of the class for KABL could be somehow modified, as I proposed in my comment for line 209. The filtering could be of the "real-time" if it is done relatively the past classifications.**

Yes, the comment 21 was full of good ideas to be put in the prospects. We will add the use of past classification to our answer to this comment.
As a matter of fact, the sensitivity analysis revealed that concatenating previous profiles do not solve these oscillations. Therefore, it is really the previous output of the classification that should be used in future filtering.

**30. Line 422: "5.6 KABL is "trainingless"" – I suggest that KABL could be used also by an expert to simplify the learning stage of supervised ML.**

As we understand this comment, the referee suggests to make first an unsupervised classification (with KABL), correct it manually and then use it as a reference to train a supervised classifier (as ADABL). This a very interesting strategy to reduce the burden of supervision in ML methods, even beyond the only question of boundary layer height estimation. For example (still close to the topic), this method is currently under experimentation to make boundary layer classification: https://github.com/ThomasRieutord/bl-classification
However we would like to emphasis that the manual correction between unsupervised and supervised classification can hardly be by-passed. First, if the result of unsupervised learning can be used as a reference, why use a supervised model after? Second, unsupervised learning tells which classes are different but not which class is what. The identification of the classes must be done by a human expert or a reliable (physically-based) strategy.

**31. Line 417: ". . .strategies . . . for the training of ADABL. . ." – To decrease the sensitivity to "idealized" diurnal cycle of the BLH, I suggest trying to exclude the time predictor in ADABL.**

Yes, this is an idea. Furthermore, a sensitivity analysis, as was done for KABL, would be very helpful to

know better how to correctly set ADABL.

**Technical corrections**

Thanks, will be corrected in the submitted version

**32. Line 13: "Atmospheric boundary layer concentrates many scientific challenges (small scale flows, turbulence...) and with high impacts due to its position of the inter-face between ground and atmosphere." - awkward English, please correct.**

**33. Line 27: "(clouds, residual layers..)" -> "(clouds, residual layers. . .)".**

**34. Line 88: "SIRTA" - Please decrypt the abbreviation.**

**35. Line 92: "at 11:15 AM and PM" – Please utilize the same notation for the time here and further. I suggest the UTC format.**

**36. Line 94: Please explain what the theta is. Is it the potential temperature?**

**37. Line 127: I suggest inserting a comma after "height above ground".**

**38. Line 129: "[[1,N]]"- What does double brackets means? Please explain.**

**39. Line 145: "tree" -> "trees".**

**40. Line 146: "m=200" -> "M=200" + upper case in sum limit**

**41. Line 154, 157, fig. 4: "top"->"left", "bottom"->"right".**

**42. Line 208: Init parameter is not defined.**

**43. Line 208: "specified in algo" -> "specified by algo parameter"**

**44. Line 270: "(0.20)" Is it the software version? Please specify.**

**45. Figure 10: Please introduce the INDUS abbreviation.**

---

## Author Response (AR2)

**Article AMT-2020-78 – Response of Referee 2's comments**

**Iteration #2**

*Note:* every response is followed by corresponding changes (CC) in the manuscript. Changes are referenced with lines numbers from the marked-up version to make them easier to identify. Changes visible in marked-up version that are not referenced as referee's response are for English language improvement.

**General comments**

In the article "Mixing height derivation from aerosol lidar using machine learning: KABL and ADABL algorithms", two machine-learning algorithms presented for the definition of the atmospheric boundary layer height, which is a principal parameter for the atmospheric modeling and air pollution dispersion.

I appreciate author answers to questions and suggestions of reviewers and I found that the second revision of the manuscript is definitely better than the first revision.

The authors are pleased to read that their efforts were appreciated and they would like to thank the reviewer for participating to the improvement of the manuscript by making thoughtful comments.

**Detailed comments**

Nevertheless, I found that the structure of the article could still be improved. For example,

1. Line 223: "– classif_score: The internal score used to automatically choose the number of clusters (only used when n_clusters='auto')." I suggest adding a reference to a following appropriate section.

The authors are not sure to correctly understand this comment because a reference to the section 3.4, where the classification scores are described, is already given line 250. These scores are also listed in the Table 1, therefore a reference to the Table 1 was added too.

*Corresponding Changes (CC): add a reference to Table 1 next to the reference to Sect. 3.4 at line 250 (marked-up version).*

2. Line 203: "1. Initialization: K centroids". A question arises how this K is defined.

Yes, the definition of K is added in the sentence.

*CC: add "where $K$ is the number of clusters specified by the user", line 227.*

There are a few problems with equations and definitions:

3. Lines 270, 271 after equations, E and sigma are not defined; E is presented lately in line 312. Even if the expectation and the standard deviation are well known, they should be once defined in the manuscript.

The definition of E[.] and \sigma(.) are added in the text.

*CC: add "and E[.] denoting mathematical expectation, \sigma(.) denoting the standard deviation", line 302.*

4. Line 311: Please verify the equation as an expectation is a constant, and the variance of a constant is always zero.

The expectation is a constant, but the conditional expectation with respect to random variable (or a σ-algebra) is a random variable, by definition. Therefore, the variance of the conditional expectation is not always zero.

*CC: none.*

5. Line 284: N and K are not defined in the equation

The corresponding paragraph was heavily edited. The definition of N and K were added in the text at this occasion.

*CC: definition of N and K added line 319-320.*

**Other questions:**

6. Line 188: "The accuracy was estimated by group K-fold, ... ", do you mean K-fold cross-validation? In my opinion, the cross-validation is a key word.

Yes, adding the key word "cross-validation" makes the sentence clearer, therefore it will be added. Group K-fold is a variant of K-fold where the K parts of the dataset are taken among Kg>K groups and the groups are specified by the user. Here, groups where chunks of four consecutive hours. It was the chosen solution to ensure that the cross-validation is done properly in the time series (as previously mentioned by the referee in its previous comments).

*CC: add "cross-validation" after "K-fold", line 213.*

7. Line 192: "An independent validation set was not used here because the accuracy was only used to discriminate between the classification algorithms." Do you mean that the aim was to compare classification algorithms using cross-validation accuracy?

Yes, supervised classification algorithms were compared according to the accuracy estimated by group K-fold cross-validation.

*CC: "accuracy" → "cross-validation accuracy", line 217.*

8. Line 252: "Finally, we look for the first change in the cluster attribution, starting from the ground level." The justification of the algorithm should be somehow presented. I suggest adding an example of KABL classification as one as in previous author's answer (question 21. Line 209) to illustrate the problem.

The justification for taking the first change is the definition of the boundary layer as the "layer of atmosphere connected to the ground". However, the suggested figure is also included as the Figure 12 in the new version of the manuscript. The authors did not judge the Section 3.3 is the most appropriate to comment on that figure and preferred the Section 5.2 to do so.

*CC: add "by definition of the boundary layer as the layer directly influenced by the ground...", line 279.*

9. Line 434: "They occur with KABL because clusters do not always have vertical persistence (some points are identified as a free atmosphere in the middle of the boundary layer)". As in the previous question, I suggest adding an example of KABL classification as one as in previous author's answer (question 21. Line 209).

The mentioned figure (question 21, line 209 of previous authors' answer to review) was added as Figure 12 in the new version of the manuscript. It is used to illustrate the problem of vertical persistence of the clusters, as well as the different options available to solve it in the current version of the code. The Sect. 5.2 was modified to include the description of this Figure.

*CC: add Figure 12 and heavy edition of the Section 5.2.*

10. Line 301: "The most relevant configurations were retained and tested on the two-year dataset." Do you mean the two-year radiosonde dataset?

No, as KABL apply to lidar data, we refer here to the two-year lidar dataset. However, the lidar dataset is also covered by radiosondes twice a day, therefore we did not think of precising more than "two-year dataset".

*CC: add precision "lidar dataset", line 340.*

11. Line 302: "There are height parameters..." , I believe that it is eight (8) parameters.

Yes, sorry for the typo.

*CC: remove the "h".*

**Article AMT-2020-78 – Response of Referee 3's comments**

**Iteration #2**

*Note:* every response is followed by corresponding changes (CC) in the manuscript. Changes are referenced with lines numbers from the marked-up version to make them easier to identify. Changes visible in the marked-up version that are not referenced as referee's response are for English language improvement.

**General comments:**

1. The paper needs to undergo English proofreading. For example, some small grammar mistakes exist throughout the paper, such as in line 72: "one of the six sensor". Another example, the use of punctuation also requires to be improved. Please check the words: therefore, however, and so on.

The manuscript was submitted to English proofreading and it appeared that there were many grammatical errors introduced into the text, which supports the reviewer's observation that that the document needed English editing. There were also subject-verb problems, spelling problems, missing articles, and missing punctuation. We hope this English proofreading has corrected all errors and now matches the journal's requirements on language.

*Corresponding Changes (CC) : many corrections throughout all the manuscript.*

**Some detailed comments:**

2. The motivation of using machine learning algorithms is not clearly indicated in the abstract.

The main motivation is the ability of machine learning to reproduce human expertise, which is acknowledged as the best way to derive boundary layer height. Major changes in the abstract to clearly motivate the use of machine learning algorithms. To keep it short, references to secondary elements were removed or shortened.

*CC: major changes in the abstract.*

3. Also, when performance was mentioned in the abstract (e.g. outperform, discrepancy etc.), the used metrics should be stated.

The metrics used to discriminate KABL, ADABL and the manufacturer's algorithm are RMSE (the lower the better) and Pearson correlation coefficient (the higher, the better) with respect to colocated radiosondes. The word "results" was replaced by explicit reference to these metrics.

*CC: "discrepancy in the results" → "discrepancy in terms of RMSE and correlation with RS", lines 13 and 558 (marked-up version).*

4. The abstract conclusion stating that ADABL is a promising algorithm, should be accompanied with key metrics evidences.

The RMSE values at the Trappes site were added to support this conclusion.

*CC: add for ADABL "(RMSE of 550 m at Trappes, 800 m for manufacturer)" + Kabl "(RMSE of 800m at Trappes)", lines 15-16.*

5. The motivations of the need of derivation of BLH needs to be improved. For example, Some sentences are not explained properly. For example, line 29: "Others methods are based on derivatives.". The word of "derivatives" here are not clear, even after we read the next sentences. Another example, line: 34-35, the abbreviations of STRAT and CABAM are also not expanded and explained. Also, in line 38: "PathfinderTURB".

Additional end-users of the BLH parameter (physical processes research) where included in the introduction. Joined with the effort to clarify the paragraph that follows, we hope the motivation of the need of BLH derivation is improved enough. We acknowledge that the use of "derivative" is misleading here. It refers to the calculation of the derivative function, while, at many occasions in the paper, the word "derive" is used as a synonym of "estimate". Therefore, this specific use of the word derivative has been clarified. We did not find explicitly any specific details about *pathfinderTURB* name in its reference article. It seems that it's just the name of the new and extended version of the *pathfinder* algorithm.

*CC : add a sentence at lines 26-28; expand algorithm acronyms for STRAT and CABAM, line 47-49; "derivative" → "calculation of the derivative function", line 41.*

6. The paragraph starting by line 40 should be used to explain AI first briefly, as some readers in journal may not be familiar with AI definition.

We agree with this comment and we added the definition of AI in the sentence introducing the acronym.

*CC: "(AI)" → "(AI), as the set of techniques aiming to reproduce...", line 55.*

7. The authors need to describe clearly supervised and unsupervised learning concept, which can be done between lines and 55.

An additional sentence on the distinction between supervised and unsupervised algorithms was included. More generally, the introduction of acronyms was entirely proofread.

*CC: add the sentence "Algorithms classifying from a reference dataset (as ADABL) are called supervised algorithms while algorithms classifying without reference dataset (as KABL) are called unsupervised.", lines 70-72.*

8. Some words are not known by wider audiences. For examples, line 95. The word row2l1 is not known well by the readers.

*raw2l1* is the name of the software used to prepare the lidar data. We re-organized the sentence to make it clearer, also we changed the word "routine" by "software" which seems more explicit. The use of italic is reserved to software which are named with a common name.

*CC: we set footnote anchor at the first occurrence of raw2l1 and put what remains of the sentence in the footnote, to clarify "was developped" is replaced by "is maintained", line 115.*

9. There are numerous supervised learning algorithms, the authors did not state clearly why boosting algorithms were chosen. Instead, the authors just stated that "the boosting algorithms are a very powerful family of algorithm".

Several algorithms were put in competition to maximize accuracy (estimated by group K-fold cross-validation, as explained in Sect. 3.1.3). The results of the competition (not shown in the paper) are the following:

[Figure]

AdaBoost has the best accuracy in average on all cross-validation splits. Its computing time is, although the highest, still affordable. Therefore it was chosen to make the supervised classification.

*CC: add a sentence to precise that several algorithms were tested, line 154-155.*

10. The KABL flowchart needs to be shown as a pseudo-code. The reviewer is also wondering, why the flowchart was made only for KABL, not for AdaBoost algorithm. If the paper attempts to compare both algorithms. They should be described fairly.

The Figure 6 was modified to highlight the common steps between KABL and ADABL (all but apply_algo). As suggested, a pseudo-code was added in stead of the "Data" information. In the pseudo-code, we focused on the parameters of KABL because they are subject to a sensitivity analysis in the Sect. 4.1. The many abbreviations were also precised in the caption.

*CC: new Figure 6*

11. The notations are not fully described in equations (1) and (2).

The subsections 3.4.1 and 3.4.2 where re-organized in order to clearly introduce all notations, number all equations and improve coherence between these two subsections.

*CC: editing of Sect. 3.4.1 and 3.4.2.*

12. The equations in subsections 3.4.2 are not numbered. The notations must also be explained properly. The reviewer understands that these are from textbook, but if the authors decided to include those in the manuscript. The notations must be clearly described.

Same as previous comment.

*CC: editing of Sect. 3.4.1 and 3.4.2.*

13. Table 1 has no explanation. All of items inside the table must be described.

The caption of Table 1 is above the table. It has been expanded with a reference to the paragraph were the items inside are described.

*CC: caption of Table 1, line 336.*

14. The fonts in Figure 9 are not readable.

The fonts of Figure 9 were enlarged.

*CC: new Figure 9.*

15. Caption for Figure 11 needs to be expanded a bit.

The caption of the Figure 11 was expanded with a description of the different elements in the figure and the expansion of the acronym "INDUS", which was not previously introduced.

*CC: caption of Figure 11, line 461.*

16. Any idea, what is manufacturer algorithm does mathematically?

The reference given by the manufacturer is Brooks, 2003. We mention it in the manuscript at line 409 (marked-up version). From what we understand, the basis of the manufacturer algorithm is a Haar wavelet covariance transform method. It consists in calculating the convolution of the lidar backscatter signal with a Haar wavelet. The maximum of the convolution is identified as the boundary layer top. We do not have any specific details about the implementation of the method. For example, we do not know how the dilation parameter of the wavelet is chosen.

*CC: none*

17. Please state future direction of the research and this work.

Future directions are given in the Section 5 which was renamed "Discussion and prospects" to make it more explicit. We chose to put the discussion and future directions before the conclusion because the conclusion includes material from the discussion and we would like to keep our conclusion concise.

*CC: change of Section 5 title to "Discussion and prospects"*

---

## Author Response (AR3)

**Article AMT-2020-78 – Response of Referee 3's comments**

**Iteration #3**

**General comment**

The authors revised significantly the manuscript based on the previous comments and suggestions. To ensure the paper quality to be high, the reviewer advised the authors if the manuscript can be checked once more in term of English grammar and presentation.

The authors made a series of little modifications in order to improve the language which are highlighted in the marked-up version. They thank the referee for the time spent on the review.

**Article AMT-2020-78 - Response of Referee 4's comments**

**Iteration #3**

*Note:* every response is followed by corresponding changes (CC) in the manuscript. Changes are referenced with lines numbers from the marked-up version to make them easier to identify. Changes visible in marked-up version that are not referenced as referee's response are for English language improvement.

**General comments**

The manuscript presents a comparison of two machine learning algorithms for boundary layer height detection using MicroMPL measurements. The optimal configurations for the algorithms have been determined using a sensitivity analysis. The performances of both methods have been studied for a 2-year database and for a case study, comparing them with independent radiosonde estimates as well as with the estimates given by the instrument manufacturer. The manuscript presents high scientific level, significance and potential for application of the work. In my opinion, the manuscript is well written, the different parts are clearly presented, and the approach is technically well justified and validated. The abstract is accurate and concise, the introduction is accessible, presents the topic background in a proper form and is followed by a clear presentation of the work done. Previous works on the subject are properly cited and the new points are clearly indicated. The methodology is properly explained, as it is expected in this kind of studies. The results and discussion sections are well structured and their content is relevant to properly analyze the proposed algorithms. Therefore, I propose that this article is accepted for publication, after improving some minor aspects that, to my view, will make the work more robust.

The authors are pleased to read such positive opinion about their work and they are thankful to the reviewer for the time spent on a careful reading.

**Main aspects**

1. The concept of "mixing height" only appears in the title and in the conclusions. However, this concept is not properly discussed or explained, and the term "Boundary layer height (BLH)" is the one that is mentioned throughout the manuscript. I agree to use this more general name (BLH) in the present work, as I understand that the aim is not to separate or detect different phenomena within BL (e.g. mixing, stable nocturnal layers, etc.). Therefore, I think the term "mixing" should be avoided, in special for the title.

The title and the conclusions were changed accordingly.

*Corresponding changes (CC)*: title and line 547 (marked-up version) "mixing height" changed for "boundary layer height".

2. To my opinion, the training of the manuscript using estimation "by hand" (as it is stated in line 170 of latest manuscript version, "On days where the boundary layer is easily visible to a human expert, the top of the boundary layer can be drawn by hand") is well justified in the view of the explanations given in the following paragraph (lines 173-183). However, in order to avoid any doubt on the justification of this estimation, I recommend: 1) to include in the introduction some brief explanation with references on the relevant aspects of BL structure that are later mentioned in lines 173-183 (i.e. stable layer, etc) and 2) to give more quantitative information in such description (e.g. "a stable boundary layer is present near the ground during the night, as it is observed by the higher signal intensity due to higher aerosol concentration..." in line 175). These corrections will also help justifying other statements given in the manuscript, e.g. "The boundary layer was clearly visible and had nearly all the features of the conceptual image" in lines 413-414.

The authors understand from this comment that the manuscript would be clearer with additional details about the terms "mixed layers" and "stable layers". However, the question of defining these terms is deep –especially seen from a measurement point of view– and is out of the manuscript's purpose. For example, the stable layer will be seen very differently with an aerosol lidar than with a radiometer, and it will be very different in a narrow valley than in large plains. Any description of the stable layer not mentioning these aspects would be incomplete and would raise legitimate critics. The authors believe that the given descriptions are understandable for AMT's readers. This choice is coherent with the newly changed title that no longer mention "mixing layer".

CC: none.

3. Nighttime retrievals have been sometimes removed from the study, in particular for the overall comparison with RS (as it is stated in line 362) and for the monthly average (line 386). However, this is not well justified in any of the cases, and actually in section 2.2 it is said that the BLH was calculated using bulk Richardson number method for nighttime because it is a good method according to literature.

The boundary layer at night is much more complex than during the day. Therefore we judged that nighttime estimates were not "well-defined cases" (line 482). This choice is debatable but it has the two following advantages.

- If a method do not work with well-defined cases, it cannot provide continuous estimation of the BLH. Therefore, assessing the new algorithms in well-defined cases appeared to the authors as a good first step.
- The overall comparison results are intended to be compared with previous studies, therefore we better reproduce similar conditions.

However, BLH-RS estimates at night were used elsewhere (e.g. in the Figure 10 b and d). For monthly average, the nighttime estimates were drawn, but not shown in the manuscript because they did not bring enough new information compare to the other figures.

CC: none.

**Other corrections**

4. Lines 180-181: "The mixed layer started to develop at 09:00 UTC...". In the view of Figure 4 (right), this development seems to start around 8 UTC actually.

Yes, the hour is corrected.

**5. Both in Figure 4 and Figure 11, an indication on sunrise and sunset times would be helpful.**

Sunrise and sunset times were not used by the human experts nor by machine learning algorithms, therefore the authors did not think this would be relevant in Figure 4. However, we do agree that this is helpful in Figure 11 in order to correctly criticize the results.

*CC:* Figure 11 and caption.

**6. Lines 190-192: lower case letters should be used after semicolon.**

Yes, it is corrected.

7. Line 339: "optimal" instead of "most optimal". This word already includes the superlative meaning.

Yes, "most" is removed.

**8. Caption of Figure 9: an indication of what INDUS means should be given.**

An additional sentence is accordingly added in the caption.